



# TROPOMI Level 3 tropospheric NO$_2$ Dataset with Advanced Uncertainty Analysis from the ESA CCI+ ECV Precursor Project

Isolde Glissenaar[1], Klaas Folkert Boersma[1,2], Isidora Anglou[1], Pieter Rijsdijk[1,3,4], Tijl Verhoelst[5], Steven Compernolle[5], Gaia Pinardi[5], Jean-Christopher Lambert[5], Michel Van Roozendael[5], and Henk Eskes[1]

[1]Royal Netherlands Meteorological Institute (KNMI), Satellite Observations department, De Bilt, The Netherlands
[2]Wageningen University, Meteorology and Air Quality group, Wageningen, The Netherlands
[3]Vrije Universiteit, Department of Earth Sciences, Amsterdam, The Netherlands
[4]SRON Netherlands Institute for Space Research, Leiden, The Netherlands
[5]Royal Belgian Institute for Space Aeronomy (BIRA-IASB), Brussels, Belgium

**Correspondence:** Klaas Folkert Boersma (folkert.boersma@knmi.nl)

**Abstract.**

We introduce the new ESA Climate Change Initiative TROPOspheric Monitoring Instrument (TROPOMI) global monthly Level 3 (L3) dataset of tropospheric nitrogen dioxide (NO$_2$) for May 2018 to December 2021. The dataset provides spatiotemporally averaged tropospheric (NO$_2$) columns, associated averaging kernels and L3 uncertainties at spatial resolutions of 0.2°, 0.5°, and 1.0° on a monthly timescale (https://doi.org/10.21944/CCI-NO2-TROPOMI-L3). To improve our understanding of what fraction of the L2 uncertainty cancels when averaging over space or time (i.e. the random component of the L2 uncertainty) and what fraction persists despite the averaging (systematic component), we first determine spatial and temporal error correlations for all sources of uncertainty in the L2 retrieval. Spatial error correlations arise from the stratosphere-troposphere correction, and from coarse-gridded albedo climatologies used in the L2 air mass factor calculation. We find the temporal error correlation in both the stratospheric uncertainty and the air-mass factor uncertainty to be 30%. Using these estimates, the L3 uncertainty budget has been established for every grid cell based on input L2 uncertainties and new methods to estimate spatial and temporal representativeness uncertainties and to propagate measurement uncertainties through space and time. The total relative uncertainty in the resulting Level 3 dataset is in the range of 15-20% in polluted areas, which is significantly lower than in separate Level 2 orbit retrievals, and brings the tropospheric NO$_2$ data to within the GCOS 'goal' and 'breakthrough' requirements. Validation of the (sub-)columns confirms better correlation and reduced dispersion in the differences between satellite and ground-based reference data for the L3 data w.r.t. the underlying L2, albeit with a more pronounced negative bias in the tropospheric columns at pollution hot spots, most probably related to stronger spatial smearing.

## 1 Introduction

Long-term monitoring of nitrogen oxides (NO$_x$ = NO + NO$_2$) in the atmosphere is crucial for several reasons. Nitrogen oxides are harmful air pollutants and long-term exposure is causally linked to chronic respiratory diseases and mortality in humans (Faustini et al., 2014; Fischer et al., 2015). When nitrogen dioxide (NO$_2$) is oxidized in the atmosphere, it forms nitric acid





(HNO$_3$), which readily dissolves in airborne droplets, and subsequently comes down as acid rain. Excessive deposition of HNO$_3$ has been shown to lead to adverse feedbacks on land- and water ecosystems (Tan et al., 2020), and changes in N-deposition influence ecosystem carbon sinks, affecting the biosphere's capacity to capture carbon from the atmosphere in the long run (Liu et al., 2022). Moreover, the monitoring of NO$_x$ concentrations helps to identify major sources of CO$_2$, because both species are co-emitted upon combustion of fossil fuels by vehicles, industrial activities, and power plants (e.g. Zhang et al., 2023). Finally, NO$_x$ play a significant role in the formation of ozone smog (Zhang et al., 2021) and fine particulate matter (Zhang et al., 2015), and both these secondary pollutants further harm human health and the environment. The importance of compounds and their indirect impact on methane, ozone, and aerosol is an important reason why tropospheric columns NO$_2$ along with other trace gases have been selected as the so-called Essential Climate Variable (ECV) Precursors for Aerosols and Ozone (Zemp et al., 2022).

To allow for policy development and air pollution and climate change assessments, inter-annual changes and trends in recent decades need to be calculated. For this, long-term, robust, sustainable, and scientifically sound Climate Data Records (CDRs) are needed to provide trustworthy information on how, where, and to what extent nitrogen oxide concentrations are changing. Atmospheric monitoring of NO$_x$ involves application of in situ measurement techniques from ground-based or airborne platforms, but these measurement techniques are limited in their spatio-temporal coverage. Satellite remote sensing with UV/Vis sensors, by contrast, provides global measurements of tropospheric NO$_2$ columns since the mid-nineties. The wide spatial coverage and continuity of satellite measurements make them fit for purpose for climate monitoring.

Level 2 (L2) retrieval algorithms to derive NO$_2$ columns from raw satellite measurements (Level 1 data) have received a lot of attention (e.g. Van Geffen et al., 2022). In comparison, Level 3 (L3) data, spatially and/or temporally averaged products on a consistent grid derived from L2 data (e.g. Wei et al., 2022), have been less considered by the scientific community, despite the relevance of L3 data for model evaluation (Visser et al., 2019; Eskes et al., 2024), data assimilation (Inness et al., 2019; Sekiya et al., 2022), and climate (trend) studies (e.g. Zara et al., 2021). This emphasis on L2 retrieval algorithms is understandable, as it reflects the need to ensure the foundational accuracy of the L2 data, which is the starting point for creating high-quality L3 products. However, the quality of L3 products does not solely depend on reliable input data, but also requires a good understanding of best methods for averaging both spatially and temporally, and assessing the propagation of existing measurement uncertainties and the quantification of additional sampling uncertainties.

Including rigorous uncertainty information in L3 CDRs is important to support the application of the data (Merchant et al., 2017). This is necessary to avoid misinterpretation of artefacts arising from system limitations as real geophysical changes or trends (e.g Labzovskii et al., 2024), for modellers to get confidence in discriminating model–data discrepancies that un-ambiguously indicate model deficiencies from those where observational errors are significant, and to contribute appropriate weighting to different observations in data assimilation and reanalyses (Merchant et al., 2017). An accurate quantification of L3 uncertainties includes both the assessment of the magnitude of error sources, as well as a propagation of these uncertainties to the L3 data product, including a treatment of spatio-temporal error correlations between individual satellite observations and of aspects of spatiotemporal representativity.

Measurement uncertainties are often assumed to be either fully systematic or fully random in space and time when determining gridded datasets (e.g., Wenig et al., 2008; Chan et al., 2023). In reality, there will be spatial and temporal correlations between multiple error sources, depending on length and time scales. There are datasets where uncertainties are determined using partial error correlations, but until now the correlation coefficients used have been determined using expert opinion, introducing subjectivity (Miyazaki et al., 2012; Boersma et al., 2016, 2018). A more quantitative treatment of spatial error correlations and representativeness errors in case of incomplete coverage of a grid cell has been presented in Rijsdijk et al. (2024). In this paper we extend this approach to temporally-averaged L3 products (gridded monthly means). This brings us to the following research questions for this study: (1) Can we improve our understanding of how uncertainties in satellite-derived tropospheric $NO_2$ columns are correlated in space and in time? (2) Can we use this understanding of correlations in uncertainties to better characterise how these uncertainties propagate in to gridded, monthly mean uncertainty estimates? (3) What is the monthly mean uncertainty budget for TROPOMI $NO_2$ L3 data, and how do these L3 uncertainties vary in space and time? (4) To what extent does the validation of with independent reference measurements help to assess the quality and fitness-for-purpose of the TROPOMI $NO_2$ L3 data? We present the ESA CCI+ L3 TROPOMI atmospheric $NO_2$ dataset with a thorough assessment of the L3 uncertainty, combining measurement uncertainties, sampling uncertainties, and a proper assessment of local error correlations for the uncertainty propagation. This includes, for the first time, an empirical quantification of correlation coefficients for multiple error sources.

## 2 Instrument and dataset

### 2.1 TROPOMI instrument

The TROPOspheric Monitoring Instrument (TROPOMI) (Veefkind et al., 2012) provides observations of tropospheric $NO_2$ columns with daily global coverage at spatial resolution of 7 x 3.5 $km^2$ and, since 6 August 2019, of 5.5 x 3.5 $km^2$ at nadir. TROPOMI is aboard the European Space Agency (ESA) Sentinel-5 Precursor (S5P) satellite, which was launched on 13 October 2017 and has been providing nominal observations since May 2018. The near-polar sun-synchronous orbit provides afternoon observations with an equator local overpass time of 13:30 h and a nearly daily global coverage.

### 2.2 TROPOMI Level 2 tropospheric $NO_2$ columns and uncertainties

The starting point for the generation of L3 data are the L2 TROPOMI satellite observations of the $NO_2$ tropospheric column on an orbital basis (Copernicus Sentinel-5P, 2021). The aim of the ESA CCI+ project is to generate long-term climate data records and therefore we use the v2.3.1 L2 data, which are the most consistent with the OMI (QA4ECV v1.1 product (Boersma et al., 2018)) because these OMI and TROPOMI data products are based on strongly consistent algorithms that use the same OMI surface albedo climatology (the so-called MINLER (Kleipool et al., 2008)), which allows for better merging of the datasets. Still, the methods described in this paper are applicable on the operational dataset or any other version of the L2 data as well. TROPOMI retrievals with qa-values of >0.75 were used, which corresponds to good-quality retrievals over (nearly) cloud-free





scenes. Retrievals made in the descending part of the orbit are removed because high latitude retrievals in the descending part of the orbit are not being used in the stratospheric correction procedure, leading to high-biased stratospheric $NO_2$ columns, and, on average, negative tropospheric $NO_2$ columns for the descending part of the orbit (see Appendix A). A small bug with

respect to the quality assurance (qa) value over snow and ice present in this version was corrected (see Eskes and Eichmann (2023)).

The $NO_2$ retrieval procedure consists of three steps, the spectral fitting, the stratospheric correction, and the conversion of the tropospheric slant column density into a tropospheric vertical column density using the air-mass factor. Each of these three steps introduces potential errors and contributes to the overall uncertainty. The single tropospheric column uncertainty ($\sigma_i$) is

quantified as in Boersma et al. (2004): $\sigma_i = \sqrt{\left(\sigma_{N_s}\right)^2 + \left(\sigma_{N_s^{strat}}\right)^2 + \left(\sigma_{M^{tr}}\right)^2}$

where the tropospheric column uncertainty sources are

$$\sigma_{N_s} = \frac{\sigma'_{N_s}}{M^{tr}} \quad ; \quad \sigma_{N_s^{strat}} = \frac{\sigma'_{N_s^{strat}}}{M^{tr}} \quad ; \quad \sigma_{M^{tr}} = \frac{(N_s - N_s^{strat}) \cdot \sigma'_{M^{tr}}}{(M^{tr})^2}$$

with $N_s$ the slant column density, $N_s^{strat}$ the stratospheric slant column density, and $M^{tr}$ the tropospheric air-mass factor and $\sigma'_{N_s}$, $\sigma'_{N_s^{strat}}$, and $\sigma'_{M^{tr}}$ their respective uncertainties.

The slant column density (SCD) uncertainty $\sigma'_{N_s}$ is estimated on a per-pixel basis during the spectral fitting and is obtained from the diagonal of the covariance matrix of the standard errors (Van Geffen et al., 2020) and has typical values of $\sim 0.6 \times 10^{15}$ molecules cm$^{-2}$ (Van Geffen et al., 2020, 2022) which compares well with estimates provided by alternative processors.

The stratospheric slant column uncertainty $\sigma'_{N_s^{strat}}$ is based on a global statistical analysis of results from the data assimilation procedure used to separate the tropospheric and stratospheric columns. The data assimilation procedure uses observational

start fields and TM5-MP 24-hr forecast stratospheric $NO_2$ fields (after modelled transport and chemistry). The difference between modelled forecasts and the actual observations (O-F) over unpolluted scenes is taken as an upper limit of the uncertainty in stratospheric $NO_2$ columns and has a statistical global-mean value of $0.2 \times 10^{15}$ molecules cm$^{-2}$ (Dirksen et al., 2011), which is applied in the L2 algorithm. Here, we apply a more detailed latitude- and time-dependent L2 stratospheric uncertainty as derived by Rijsdijk et al. (2024). A recent comparison of stratospheric $NO_2$ columns obtained with data assimilation

in TM5-MP and the STREAM approach (Beirle et al., 2016), an alternative method to separate the stratosphere and troposphere columns, for GOME-2A showed consistency to within $0.2 \times 10^{15}$ molecules cm$^{-2}$ (Richter et al., 2024). Similar results were found in a comparison of stratospheric $NO_2$ columns from the data assimilation method and STREAM for the Ozone Monitoring Instrument (OMI) of the QA4ECV data set (Boersma et al., 2018; Compernolle et al., 2020).

The air-mass factor uncertainty $\sigma'_{M^{tr}}$ consists of uncertainty contributions from the cloud pressure, cloud fraction, surface

albedo, and a priori $NO_2$ profile shape. The theoretical error propagation framework in Boersma et al. (2004) is used to estimate the overall AMF uncertainty. The overall tropospheric AMF uncertainties are estimated to be 30-50% (Boersma et al., 2018) for individual retrievals.

The overall uncertainty for an individual retrieval therefore depends on details in the retrieval and is pixel specific. Over oceans and remote areas, with low tropospheric vertical columns, the relative overall uncertainty is typically more than 100%

and is dominated by uncertainty in the spectral fit ($\sigma_{N_s}$) and the stratospheric column ($\sigma_{N_s^{strat}}$) (van Geffen et al., 2022b). For





more polluted regions over continental areas, the relative uncertainty reduces to 25-50% and is dominated by uncertainty in the tropospheric air mass factor ($\sigma_{M^{tr}}$) (van Geffen et al., 2022b). The overall uncertainty for individual TROPOMI tropospheric column NO$_2$ retrievals is sometimes approximated as: $\sigma_i \approx 0.5 \times 10^{15}$ molecules cm$^{-2}$ + [0.2 to 0.5] $\times N_v^{trop}$ (van Geffen et al., 2022b).

## 3   Methodology

### 3.1  L3 algorithm overview

Our starting point is individual L2 tropospheric column retrievals with a retrieved column value $x_i$ (molecules cm$^{-2}$) and the associated individual retrieval uncertainty $\sigma_i$ (molecules cm$^{-2}$). To compute L3 spatial and temporal gridded means, we developed a two-step procedure, in which we:

1. calculate spatially averaged (gridded) column values ($x_{o,t}$) – where $t$ stands for an instantaneous column value not averaged in time – along with their associated L2 measurement uncertainties ($\sigma_{m,s}$) and spatial representativeness uncertainty ($\sigma_{rs}$), and then

2. provide temporal averaged estimates of the column values ($\bar{x}$) and their total uncertainty ($\bar{\sigma}_{total}$), including associated temporal representativeness uncertainty ($\bar{\sigma}_{rt}$) and measurement uncertainties ($\bar{\sigma}_m$).

Figure 1 provides a schematic picture of the above procedure. After collecting the individual values of the retrieved column and its uncertainty ($x_i$, $\sigma_i$), their spatially averaged (instantaneous) counterparts ($x_{o,t}$, $\sigma_{o,t}$) are calculated for a L3 grid cell. Then, in step 2, these spatial averages are aggregated over time and averaged, leading to the desired L3 product. This two-step method allows us to assess and apply different error correlation factors when propagating uncertainties spatially and temporally.

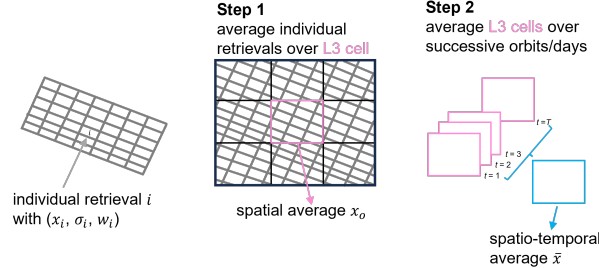

**Figure 1.** Schematic example of the procedure to calculate spatio-temporal (gridded) averages of retrieved columns and their associated uncertainties. The irregular grey rectangles indicate an ensemble of individual L2 retrievals in a satellite orbit, the black rectangle shows a regular grid to which the individual retrievals are averaged and gridded in Step 1. In Step 2, this procedure is repeated to obtain multiple spatial averages along with their associated uncertainties, which are then averaged over a period to produce a temporal mean and its associated L3 uncertainty.





## 3.2 Spatial averaging

### 3.2.1 Averaging of variables

Step 1 concerns the spatial averaging of valid retrieved values of the individual pixels ($x_i$) with weights ($w_i$). Following
Miyazaki et al. (2012); Boersma et al. (2016); Rijsdijk et al. (2024) the weight $w_i$ of each measurement is taken equal to the
spatial overlap area between the footprint of observation $i$ and the L3 grid cell. This tiling approach results in the following
estimate of the gridded L3 column,

$$x_{o,t} = \frac{\sum_{i=1}^{N}(w_i x_i)}{\sum_{i=1}^{N} w_i} \quad (1)$$

with $N$ the number of valid L2 observations in the L3 cell.

The spatial average is determined on a (regular) grid on a per-orbit basis. This is possible for any given grid, we provide
spatial the here created dataset at resolutions 0.2ºx0.2º, 0.5ºx0.5º, and 1ºx1º. The spatial average for a grid cell is only
determined if the combined valid individual retrievals $x_i$ cover at least 30% of the grid cell, to avoid unrealistic uncertainty
estimates in small data samples, following Rijsdijk et al. (2024).

### 3.2.2 Uncertainty estimate

To assess the overall uncertainty ($\sigma_{o,t}$) in $x_{o,t}$ of these spatial averages we follow the procedure in Rijsdijk et al. (2024), and
combine the propagated measurement uncertainties ($\sigma_m$) and the spatial representativeness uncertainty ($\sigma_{rs}$) in quadrature:

$$\sigma_{o,t} = \sqrt{(\sigma_m)^2 + (\sigma_{rs})^2} \quad (2)$$

This reflects that the uncertainty in the spatial average is composed of propagated L2 uncertainties, and may also contain
uncertainty from incomplete coverage of the grid cell.

**Measurement uncertainty**

The measurement uncertainty $\sigma_m$ is the combined tropospheric column uncertainty from error sources ($\sigma_{m,s}$) in the L2
retrieval, including the uncertainty due to the slant column density measurements ($\sigma_{N_s}$), the uncertainty resulting from errors
in the stratospheric column ($\sigma_{N_s^{strat}}$), and the uncertainty from the air mass factor ($\sigma_{M^{tr}}$) (Boersma et al., 2004).

$$\sigma_m = \sqrt{(\sigma_{N_s})^2 + (\sigma_{N_s^{strat}})^2 + (\sigma_{M^{tr}})^2} \quad (3)$$

Note that these are the contributions of each uncertainty source to the tropospheric column uncertainty, not the source uncer-
tainties themselves. Each of these are represented by equation 4 (Sekiya et al., 2022; Rijsdijk et al., 2024) which considers that
the random component of the individual uncertainties $\sigma_i$ tends to cancel out when averaging over many observations (first term
on the right hand side), while a fraction $\phi_s$ of the individual uncertainties $\sigma_i$ persists after averaging because they come from
systematic spatially correlated contributions to the uncertainties (second term):

$$\sigma_{m,s} = \sqrt{(1-\phi_s)\frac{\sum_{i=1}^{N}(w_i^2 \sigma_{i,s}^2)}{\sum_{i=1}^{N} w_i^2} + \phi_s \frac{(\sum_{i=1}^{N}(w_i \sigma_{i,s}))^2}{\sum_{i=1}^{N} w_i^2}} \quad (4)$$





The spatial error correlation factor $\phi_s$ in equation 4 quantifies the portion of the error that is systematic and here we determine its value for each of the measurement error contributions $s$ (slant column density, stratospheric column, and air-mass factor) separately. We apply the spatial error correlations as determined by Rijsdijk et al. (2024).

The slant column error is largely uncorrelated over space as it is dominated by measurement noise (Van Geffen et al., 2020; Rijsdijk et al., 2024). There will be a systematic error component in the slant column density due to gaps in knowledge, such as offsets in absorption cross sections, inaccurate Ring coefficients in the spectral fit or the lack of a correction for vibrational Raman scattering (Richter et al., 2011; Zara et al., 2018; Rijsdijk et al., 2024). In Van Geffen et al. (2020) the slant column retrieval noise estimated over a Pacific sector was found to match the DOAS uncertainty estimate to within 15% for clear-sky pixels, suggesting systematic retrieval errors are roughly a factor 2 smaller than the random errors. However, the effect of these systematic errors is absorbed in the stratospheric column estimate (as discussed below). Therefore, the uncertainty in slant column density is assumed to be fully random ($\phi_{N_s} = 0$) (Rijsdijk et al., 2024).

In the retrieval method the stratospheric column is determined by assimilating slant column superobservations in the chemical transport model TM5-MP (Dirksen et al., 2011). The resolution of the TM5-MP model is $1^\circ \times 1^\circ$, and the horizontal correlation length scale used in the assimilation is about $500\,\mathrm{km}$. This is coarser than the spatial average grid sizes considered here. Therefore, it is assumed that the error in the stratospheric column is fully correlated in space with the L3 grid resolution of $0.2^\circ$ - $1.0^\circ$ introduced before ($\phi_{N_s^{strat}} = 1$). A spatially correlated error in the slant column would lead to an increased bias in the O-F, increasing the L2 stratospheric column uncertainty. The spatial mean of the stratospheric column uncertainty in Equation 3 will thus include the contribution from the systematic slant column uncertainty.

The AMF uncertainty is predominantly caused by uncertainties in surface albedo and cloud parameters (cloud fraction and cloud pressure). All three of these variables depend on the quality of the climatological surface albedo dataset. Rijsdijk et al. (2024) review the AMF uncertainty by comparing versions 2.3.1 and 2.4 of the $NO_2$ L2 datasets. The main difference between these two versions is that they apply different surface albedo climatologies. We argue that the differences between albedo estimates from these state-of-science climatologies are indicative of realistic albedo uncertainties. The albedo values directly affect the clear-sky AMF calculations, but also drive the retrieved values for the cloud fraction and cloud pressure (see e.g. Riess et al. (2021)). Differences between albedo values thus propagate to different values of cloud fraction and cloud pressure, which impact the cloudy-sky AMF values. Albedo differences thus impact the overall AMF calculation, and spatio-temporal characteristics in the albedo differences are expected to lead to spatio-temporal patterns in AMF differences, allowing us to analyse how the AMF uncertainty patterns are correlated in time and space. The effects of changing the surface albedo climatology on the results of the tropospheric $NO_2$ retrieval are expressed as differences between v2.3.1 and v2.4 tropospheric $NO_2$ columns. The error in the AMF is shown by Rijsdijk et al. (2024) to be partly correlated depending on the size of the grid cells ($\phi_{M^{tr}} = e^{-d/l}$, where $d$ is a distance between observations depending on the grid cell size and $l = 35$ km is a typical average correlation length over polluted regions). This spatial correlation is partly due to the low resolution of surface albedo datasets ($0.5^\circ$), but also because surface modifying conditions are often spatially extensive. For example, droughts impact the surface albedo not just in one grid cell, but typically over a larger area. Table 1 provides typical values of the spatial error correlation





coefficients for TROPOMI spatial averages at different grid resolutions.

**Table 1.** Spatial error correlation sources and their random and systematic fractions for TROPOMI NO$_2$ tropospheric column spatial averages at different spatial resolutions. For the calculation of the AMF coefficients, $\phi_{M^{tr}} = e^{-d/l}$ has been used with $l = 35$ km the typical correlation length over polluted regions (Rijsdijk et al., 2024) and $d$ the typical grid cell dimension in km.

| Spatial error correlation source | Random fraction (averages out) | Systematic fraction (persists) |
|---|---|---|
| Slant column density | $(1 - \phi_{N_s}) = 1.00$ | $\phi_{N_s} = 0.00^\dagger$ |
| Stratospheric correction | $(1 - \phi_{N_s^{str}}) = 0.00$ | $\phi_{N_s^{str}} = 1.00$ |
| AMF ($0.2° \times 0.2°$) | $(1 - \phi_{M_{tr}}) = 0.44$ | $\phi_{M_{tr}} = 0.56$ |
| AMF ($0.5° \times 0.5°$) | $(1 - \phi_{M_{tr}}) = 0.75$ | $\phi_{M_{tr}} = 0.25$ |
| AMF ($1.0° \times 1.0°$) | $(1 - \phi_{M_{tr}}) = 0.94$ | $\phi_{M_{tr}} = 0.06$ |

† Note: the systematic component of the SCD uncertainty is included in the stratospheric uncertainty.

**Spatial representativeness uncertainty**

The spatial representativeness uncertainty $\sigma_{rs}$ (molecules cm$^{-2}$) accounts for incomplete sampling (mainly due to cloud covered pixels not being included in the spatial averaging) of the cell by the observations available to calculate $x_{o,t}$ (Rijsdijk et al., 2024). If the entire area of the L3 cell is covered by valid observations then the representativeness uncertainty is 0. If only a small fraction of the L3 cell is covered, the representativeness uncertainty is equal to the standard deviation of the tropospheric columns within the cell: large for areas with strong spatial variability in $x_i$ (such as over polluted regions), and smaller for regions with similar values of $x_i$ (such as over clean, background regions).

$$\sigma_{rs} = f\sigma_{x_i} \tag{5}$$

where $\sigma_{x_i}$ is the standard deviation in retrievals within the grid cell ($x_i$). The unitless fraction $f$ is calculated as the representativeness of the retrieved observations for the fully covered grid cell:

$$f = \frac{1}{\sqrt{N_{eff}\alpha_{o,t} + 1}} \sqrt{1 - \alpha_{o,t}} \tag{6}$$

where the degree of coverage $\alpha_{o,t}$ is calculated as the fraction of the total valid pixel area ($A$, in km$^2$)

$$\alpha_{o,t} = \frac{\sum_{i=1}^{N} w_i}{A} \tag{7}$$

and $N_{eff}$ is the number of effective observations, which is dependent on the number of available observations, the sensor, the trace gas, the L3 resolution gridding to, and whether a grid cell is sensitive to systematic sampling (due to, e.g., a cloud field covering part of a grid cell) or not (Rijsdijk et al., 2024). The lower the value for $N_{eff}$, the more sensitive a region is to systematic sampling. $N_{eff}$ has been determined empirically by Rijsdijk et al. (2024), and a summary of the methods used to quantify $N_{eff}$ can be found in Appendix B.



### 3.3 Temporal averaging

#### 3.3.1 Averaging of variables

The second step in the generation of the L3 dataset consists of temporal averaging. We average the spatial-mean values $x_{o,t}$
obtained in step 1 over time (e.g. a period of one month), again in a weighted fashion to account for different degrees of repre-
sentativity ($w_{o,t} = 1 - g$). $g$ is high for grid cells with a large representativeness uncertainty and low for superobservations with
almost no representativeness uncertainty. This value depends on coverage - higher coverage results in a lower representative-
ness error - but also other factors such as sensitivity to systematic sampling. Taking a weighted average like this implies that
superobservations with a low representativeness uncertainty obtain more weight in the temporal average than superobserva-
tions with a high representativeness uncertainty. This results in the following estimate of the spatially and temporally averaged
L3 column $\bar{x}$:

$$\bar{x} = \frac{\sum_{t=1}^{T}(w_{o,t}x_{o,t})}{\sum_{t=1}^{T}w_{o,t}} \tag{8}$$

with $T$ the total number of valid superobservations in the period over which the averaging is performed.

#### 3.3.2 Uncertainty estimate

Next, the uncertainty associated with $\bar{x}$, i.e. the monthly mean L3 uncertainty, is determined. We calculate a total spatio-
temporal averaged uncertainty which combines the propagated measurement uncertainty ($\bar{\sigma}_m$, including the spatial represen-
tativeness uncertainty) and a temporal representativeness uncertainty ($\bar{\sigma}_{rt}$) in quadrature:

$$\bar{\sigma}_{total} = \sqrt{(\bar{\sigma}_m)^2 + (\bar{\sigma}_{rt})^2} \tag{9}$$

**Propagated measurement uncertainty**

The spatio-temporally averaged measurement uncertainty $\bar{\sigma}_m$ is the combined uncertainty of the measurement error sources
$\bar{\sigma}_{m,s}$ including the spatial representativeness uncertainty:

$$\bar{\sigma}_m = \sqrt{(\bar{\sigma}_{N_s})^2 + (\bar{\sigma}_{N_s^{strat}})^2 + (\bar{\sigma}_{M^{tr}})^2 + (\bar{\sigma}_{rs})^2 + (\bar{\sigma}_{apriori})^2} \tag{10}$$

The propagated slant column density uncertainty ($\bar{\sigma}_{N_s}$), stratospheric column uncertainty ($\bar{\sigma}_{N_s^{strat}}$), air-mass factor uncertainty
($\bar{\sigma}_{M^{tr}}$) and spatial representativeness uncertainty ($\bar{\sigma}_{rs}$) are each determined using:

$$\bar{\sigma}_{m,s} = \sqrt{(1-\tau_s)\frac{\sum_{t=1}^{T}(w_{o,t}^2\sigma_{m,s_t}^2)}{(\sum_{t=1}^{T}w_{o,t})^2} + \tau_s\frac{(\sum_{t=1}^{T}(w_{o,t}\sigma_{m,s_t}))^2}{(\sum_{t=1}^{T}w_{o,t})^2}} \tag{11}$$

where $\tau_s$ is the temporal correlation factor of the uncertainty sources $s$ (not the same as the spatial correlation factor applied
in equation 4). The last term in equation 10 represents the contribution from the uncertainty in the a priori profile shapes, and
is approximated as 10% of the tropospheric AMF. 10% of the AMF is deemed appropriate for the a priori profile uncertainty,





as the spatial resolution of the spatial means (0.2°x0.2° to 1°x1°) is low compared to the TROPOMI pixel resolution. Earlier
studies (gridding OMI to 0.5°) also used an estimate of 10% (Boersma et al., 2018). $\bar{\sigma}_{apriori}$ is not included in the uncertainty
calculation when using the averaging kernel in data applications, which removes the dependence on the a priori profile (Eskes
and Boersma, 2003).

The temporal correlation factor $\tau$ is determined for each of the sources of measurement uncertainty (slant column density,
stratospheric, and AMF) separately. We do so by evaluating to what extent discrepancies in the stratospheric $NO_2$ column and
in the AMF calculation vanish over time.

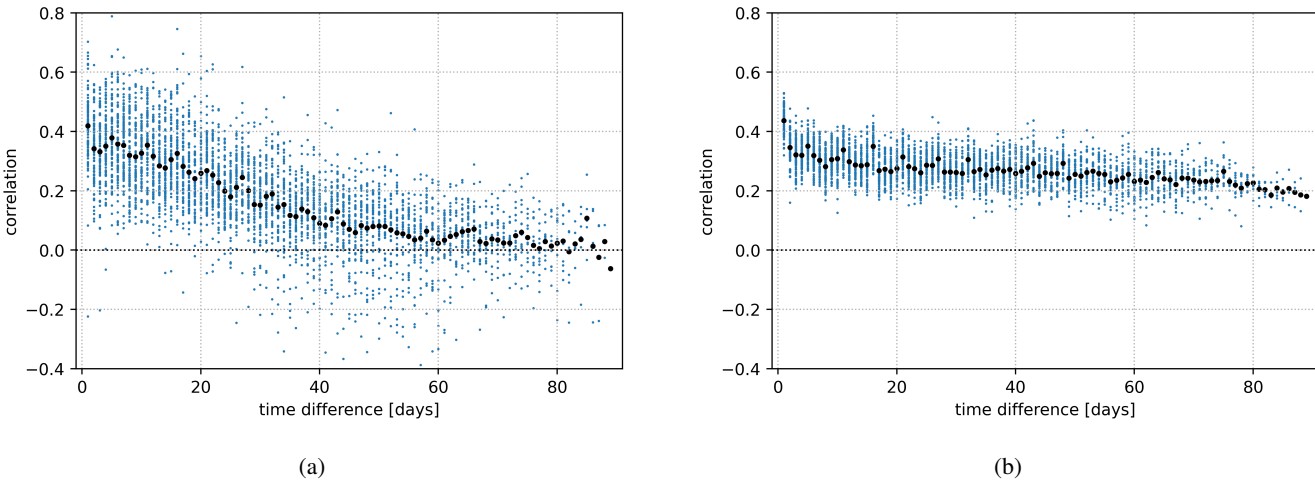

|          |          |
| :------: | :------: |
|   (a)    |   (b)    |

**Figure 2.** Temporal correlation in (a) difference of tropospheric $NO_2$ column for v2.3.1 and v2.4 for polluted areas (AMF uncertainty)
and (b) observation-forecast for unpolluted areas (stratospheric column uncertainty) between each combination of two days within the same
month for the period 1 Jan - 31 Mar 2019. Larger points are the mean of correlations with the same time difference.

Firstly, the temporal error correlation factor for the AMF uncertainty is determined. The a priori $NO_2$ profile is a large
contribution to the uncertainty in the AMF, but is shown to become irrelevant when the averaging kernel is used when com-
paring with three-dimensional model output (Eskes and Boersma, 2003). Other large sources of uncertainty in the AMF are
the effective cloud cover, the effective cloud height, and the surface albedo. All three of these variables depend on a monthly
climatological surface albedo dataset. This introduces a temporal error correlation between the daily observations.

Assuming the AMF error is fully correlated in time would result in an overestimation of the AMF uncertainty in level 3
products, as in reality the AMF error is only partly systematic and the random part averages out with many observations.
Here, we empirically evaluate the systematic character of the AMF uncertainty on different temporal scales, by determining
the degree of temporal correlation between AMF errors.





We estimate the temporal error correlation required to determine the AMF uncertainty in spatio-temporal averaged L3 data by comparing versions 2.3.1 and 2.4 of the L2 retrieval, as was done for the spatial error correlation (section 3.2.2). These versions apply different climatological surface albedo datasets, with v2.3.1 applying albedo derived from OMI and GOME-2 and v2.4

applying albedo derived from TROPOMI spectra (Tilstra et al., 2024). As the albedo is a key input for the cloud retrieval, this replacement of climatological albedo dataset also generates cloud fraction and cloud pressure retrieval differences. Because the change of albedo climatology is the largest change between the two L2 versions, the difference in tropospheric $NO_2$ between the versions should be indicative of uncertainty in tropospheric $NO_2$ resulting from the climatological surface albedo.

We determine the difference in tropospheric $NO_2$ column in $0.2° \times 0.2°$ superobservations between v2.3.1 and v2.4 L2 data

for each day in the period 1 January - 31 March and the period 1 June - 31 August 2019. For each combination of two days within the same period, we determine the Pearson correlation coefficient in tropospheric $NO_2$ for polluted areas (tropospheric column from both versions > 1.8 molecules $cm^{-2}$) (Figure 2a):

$$r_{\Delta t} = \frac{\sum_{i=1}^{I}(\Delta N_{t_1 i} - \overline{\Delta N_{t_1}})(\Delta N_{t_2 i} - \overline{\Delta N_{t_2}})}{\sqrt{\sum_{i=1}^{I}(\Delta N_{t_1 i} - \overline{\Delta N_{t_1}})^2 \sum_{i=1}^{I}(\Delta N_{t_2 i} - \overline{\Delta N_{t_2}})^2}} \tag{12}$$

where $r_{\Delta t}$ is the Pearson correlation coefficient between tropospheric $NO_2$ differences due to albedo differences and re-

trieval discrepancies on any two days in the time period (blue points in Figure 2a), $\Delta t = t_2 - t_1$ the time difference, $\Delta N_{t_1} = N_{t_1}^{v2.4} - N_{t_1}^{v2.3.1}$ and $\Delta N_{t_2} = N_{t_2}^{v2.4} - N_{t_2}^{v2.3.1}$ the difference in tropospheric $NO_2$ between v2.3.1 and v2.4 for the first day and second day respectively, and $I$ the number of valid superobservation grid cells for both days. For each 'time difference' between daily superobservations, we determine the mean correlation ($\overline{r_{\Delta t}}$, black points in Figure 2a). Then we take the mean of the $\overline{r_{\Delta t}}$ for the time differences 1-30 days. This results in $\tau = 0.29$ for 1 January to 31 March and $\tau = 0.30$ for 1 June to 31

August, suggesting that tropospheric column errors only partially persist in time. The temporal error correlation in the AMF is set to be $\tau_{M^{tr}} = 0.30$ for both the NH winter and summer period. When analysing longer time differences than a month, the correlation disappears. The method was repeated for $1° \times 1°$ superobservations, resulting in similar results, giving us confidence that the temporal AMF error correlation coefficient is not grid-size dependent.

Next, we estimated temporal correlation factors for the stratospheric uncertainty. In the retrieval of the $NO_2$ columns the observed slant column is split into its tropospheric and stratospheric parts using data-assimilation in the TM5-MP model. We can assess uncertainty and bias in this model by investigating the difference between the forecasted column and the observed column (O-F) over unpolluted regions, see e.g. Dirksen et al. (2011). We take the values for O-F for the period 1 January to 31 March 2019. For each combination of two days within the same month, we determine the Pearson correlation coeffi-

cient in stratosphere O-F for clean areas (tropospheric column < 0.6 molecules $cm^{-2}$) (Figure 2b). This is again done with equation 12 but now with $\Delta N_{t_1} = N_{t_1}^{\text{observation}} - N_{t_1}^{\text{forecast}}$ and $\Delta N_{t_2} = N_{t_2}^{\text{observation}} - N_{t_2}^{\text{forecast}}$ the difference in stratospheric $NO_2$ between observation and forecast for the first day and second day respectively. For each 'time difference' between daily O-F maps, we determine the mean correlation ($\overline{r_{\Delta t}}$). Then we take the mean of $\overline{r_{\Delta t}}$ for the time differences 1-30 days. This results in $\tau = 0.30$. The same method is carried out for the NH summer period 1 June-31 August, which showed a slightly lower

correlation coefficient of $\tau = 0.21$. We take the higher value of $\tau_{N_s^{strat}} = 0.30$ as the temporal uncertainty correlation in the





stratospheric uncertainty for both the NH winter and summer period, as a conservative uncertainty estimate. The correlation decreases slightly for longer periods and only disappears altogether for periods longer than 3 months.

The uncertainty in the slant column density is assumed to be fully uncorrelated in time ($\tau_{N_s} = 0$), following the same arguments as in the spatial uncertainty correlation (see Section 3.2.2).

   The temporal correlation coefficient of the spatial representativeness uncertainty is set at fully random ($\tau_{rs} = 0$). The representativity can be correlated through time due to the effect of persistent cloud cover, but this will be assessed separately in the temporal representativeness uncertainty.

**Table 2.** Temporal error correlation sources and their random and systematic fractions for TROPOMI $NO_2$ tropospheric column spatiotemporal averages.

| Temporal error correlation source | Random fraction (averages out) | Systematic fraction (persists) |
|---|---|---|
| Slant column density | $(1 - \tau_{N_s}) = 1.00$ | $\tau_{N_s} = 0.00^{\dagger}$ |
| Stratospheric correction | $(1 - \tau_{N_s^{str}}) = 0.70$ | $\tau_{N_s^{str}} = 0.30$ |
| AMF | $(1 - \tau_{N_{M_{tr}}}) = 0.70$ | $\tau_{N_{M_{tr}}} = 0.30$ |

† Note: the systematic component of the SCD uncertainty is included in the stratospheric uncertainty.

The monthly mean L3 $NO_2$ columns are only representative when sufficient observations are available for calculating the
means. The L3 qa-value is set to equal 1 when the data is representative as a monthly mean and equal 0 when it is not. This deviation is made using the count variable, which is the sum of the fractional coverages from the spatial averages $\alpha_{o,t}$ divided by the number of days in the month. If this value is below 0.1 (meaning less than 10% of the month is sampled), the L3 qa-value is set at 0 and the L3 $NO_2$ columns are advised not to be used for further analyses, as these would not be a good representation of the given month.


**Temporal representativeness uncertainty**

   The temporal representativeness uncertainty ($\bar{\sigma}_{rt}$) can be interpreted as the standard error of the superobservations (spatial means) used to determine the monthly mean. We state that the number of possible observations within a month is finite and a true monthly mean is obtained with at least one observation (note that the TROPOMI orbits overlap at higher latitudes, where
multiple observations per day are possible) each day. A correction factor is applied to the method of calculating the standard error to correct for this (Bondy and Zlot, 1976; Isserlis, 1918; Rijsdijk et al., 2024). This results in the following theoretical formula for the temporal representativeness uncertainty:

$$\bar{\sigma}_{rt} = \frac{\sigma_{x_{o,t}}}{\sqrt{n}} \sqrt{\frac{N-n}{N-1}} \tag{13}$$




where $N$ is the length of the month in days and $n$ is the number of days with at least one valid superobservation. The weighted standard deviation around the temporal mean $\sigma_{x_{o,t}}$ is determined using:

$$\sigma_{x_{o,t}} = \sqrt{\frac{\sum_{t=1}^{T}(w_{o,t}(x_{o,t} - \bar{x})^2)}{(T-1)\sum_{t=1}^{T} w_{o,t}}} \tag{14}$$

with $T$ the total number of valid orbits in the period over which the averaging is performed, and $w_{o,t} = 1 - f$.

We apply the method also used in Rijsdijk et al. (2024) and Appendix B to assess whether Equation 14 is suitable for assessing the temporal representativeness uncertainty. We start by taking a completely covered grid cell, which means that at least one spatial mean/superobservation was produced for this grid cell for every day of the month. The representativeness uncertainty can be quantified by comparing the mean of this completely covered grid cell to the mean of several samples (a subset of days) taken from that grid cell. We start by sampling a single day, which we use to estimate the temporal mean of the grid cell. Then we repeatedly add random days of the month and estimate the mean using the available observations. We perform multiple iterations (200) of this process on the same grid cell and find the relation between representativeness uncertainty $f_t$ and temporal coverage of the grid cell (Figure 3a). This process is then repeated for 100 grid cells in polluted areas (Figure 3b). We only take polluted grid cells into account (mean $NO_2$ tropospheric column of $\geq 2 \times 10^{15}$ molecules $cm^{-2}$), as these are expected to have the highest temporal variability and thus are more likely to be sensitive to a sampling bias. The mean of the experiments (green) is almost identical to the theoretical solution of equation 13 (blue), showing that the formula is a suitable method for quantifying the temporal representativeness uncertainty for random sampling. Similar results are found for June 2019.

As with the spatial representativeness (cloud masking one part of the grid cell), there might be a sensitivity to systematic sampling temporally. This could for example be the case if large weather systems give persistent cloud cover in a given location for part of the month. The occurrence of systematic sampling was tested for the spatial means of January and June 2019. We selected grid cells in the spatial mean that had valid superobservations for $\sim$50% of the given month. On these grid cells we applied a Wald-Wolfowitz test to determine whether the cloudiness was random or systematic (Wald and Wolfowitz, 1943). The Wald-Wolfowitz test is a non-parametric statistical test that checks a randomness hypothesis for a two-valued data sequence, in our case cloudy (no data) or not-cloudy (superobservation available, can be partly cloudy). For both January and June about two-thirds of the grid cells show systematic sampling due to continuous cloud cover for part of the month.

This shows that systematic sampling of observations for the temporal mean is relevant. Next, we will look into whether the temporal variability in the observations makes it sensitive to this systematic sampling. We repeat the method of repeated sampling of a grid cell as outlined above, but now by sampling the grid cell systematically (start by taking the observation of one random day and then repeatedly add adjacent days). Figure 3d shows that systematic sampling produces more variability of grid cells around the theoretical solution (blue line, equation 13) than with random sampling (Figure 3b), but that the mean of the experiments (green line) is close to the theoretical solution. Comparing with the spatial representativeness curve in Figure B1b shows that the experimental curves in Figure 3d are closer to the theoretical solution. The fit from equation 6 in purple does not show a better fit to the experimental mean than the theoretical solution does. This shows that the temporal mean is not



very sensitive to systematic sampling and the uncertainty can be assessed with the theoretical solution suggested in equation 13.

It should be noted that this method can only be applied to grid cells that provide coverage for all days of the month. This is essentially only true for arid areas with almost no cloud cover. The question arises how representative these regions are. We filter for relatively polluted regions ($\geq 2 \times 10^{15}$ molecules cm$^{-2}$) and find more than 100 grid cells to which we can apply this method. These are found mostly in cities in the Middle-East and western India, where January is in the dry season. We assume these polluted areas are representative for polluted areas in other parts of the world and the theoretical solution can be applied globally.

## 4 TROPOMI NO$_2$ Level 3 dataset

### 4.1 Dataset

The here presented ESA CCI+ TROPOMI Level 3 dataset is available for the period May 2018 to December 2021 at a monthly resolution. The dataset is available on a global regular grid at different spatial resolutions: 0.2°x0.2°, 0.5°x0.5°, and 1°x1°. The datasets are organised into a user-friendly and self-describing netCDF-4 format, following CF metadata conventions. The dataset represents the NO$_2$ columns at satellite overpass time ($\sim$13:30h) under mostly clear-sky conditions.

The dataset contains both the tropospheric and stratospheric vertical column density (see Appendix C for all variables). The dataset contains two estimates of the total uncertainty for the tropospheric column $\bar{\sigma}_{total}$, one including the a priori uncertainty and one version excluding the a priori uncertainty which can be used when applying the averaging kernel. A spatio-temporal average of the tropospheric averaging kernel from the L2 dataset is also available in the L3 dataset. For assessing the L3 data quality the L3 qa-value, count variable, and average cloud radiance fraction are also available. This dataset can be used for, e.g., temporal analysis, emission monitoring, data assimilation and model validation, and atmospheric chemistry studies.

### 4.2 Analysis of dataset

The results of the spatio-temporal average monthly mean and uncertainties for NO$_2$ for January and June 2019 are shown in Figure 4, with local values given in Table 3. Missing data are mainly due to polar night at high latitudes and filtering of low-quality L2 data, due to, e.g., consistent cloud or snow cover. All results presented in this section are from the 0.2°x0.2° spatial resolution dataset.

In June the tropospheric NO$_2$ columns are high in Africa due to wildfires and biomass burning in the dry season. In the urbanised areas in the Northern Hemisphere, tropospheric NO$_2$ columns have higher values during the winter season reflecting longer lifetimes, and thus we see higher values over Amsterdam and Beijing in January than in June (Table 3, Figure 6). The opposite is true for urbanised areas in the Southern Hemisphere, for example Rio de Janeiro (Table 3, Figure 6). The total uncertainty in winter when not applying the averaging kernel attributes to 21% and 17% of the total tropospheric column for Amsterdam and Beijing respectively. These percentages are 19 and 14% respectively when applying the averaging kernel. Due



**Figure 3.** Results of repeatedly sampling grid cells to calculate $\bar{\sigma}_{rt}$. (a) Repeated random sampling of 31 days of a single grid cell with values $x_{o,t}$. Daily observations in the grid cell are randomly sampled 200 times. The thin grey lines represent the difference between the sampled mean and the actual mean from individual random experiments. The green line is the mean of the samples and the blue line the theoretical result from equation 13. (b) Results of randomly sampling 100 grid cells. The grey lines are the green line from panel a. The green line (overlapping the blue theoretical result) is the mean of the results of 100 grid cells. (c) Systematic sampling of a single grid cell. (d) Results of systematically sampling 100 grid cells. Observations in all panels are from January 2019 and have a mean tropospheric column of $\geq 2 \times 10^{15}$ molecules cm$^{-2}$. Only grid cells with observation available every day are sampled.





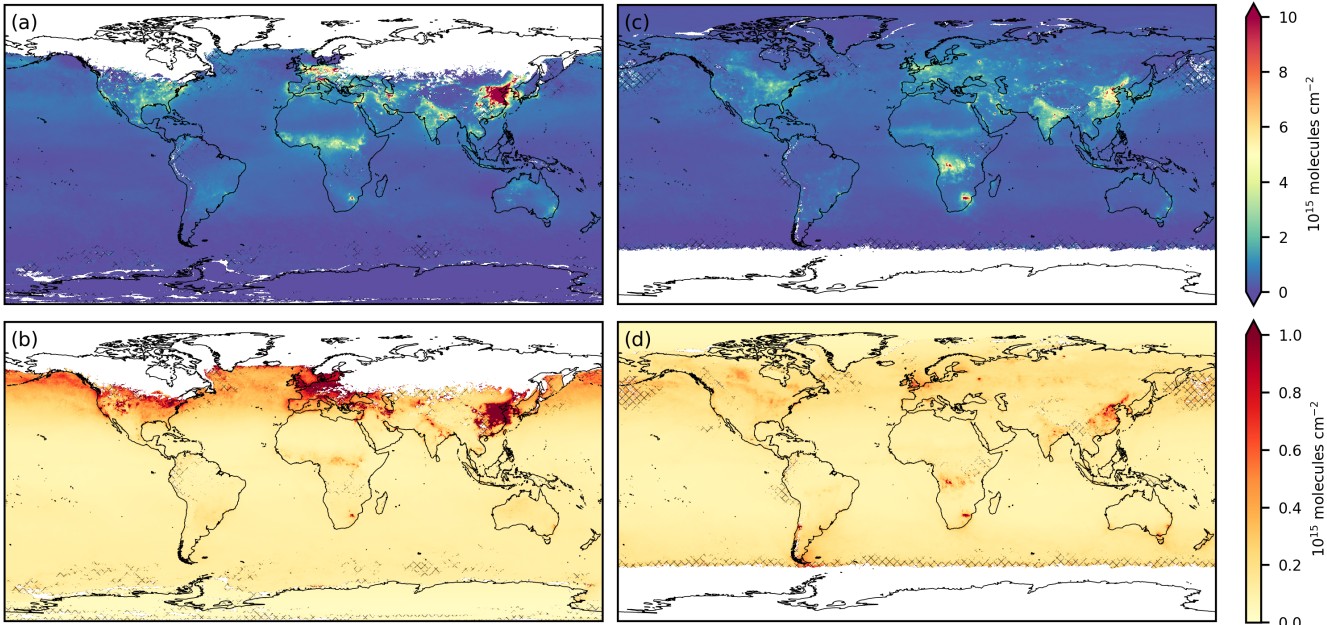

**Figure 4.** Example of results of the monthly (a) tropospheric $NO_2$ column with its (b) total uncertainty ($\bar{\sigma}_{total}$) for January 2019 and the same (c,d) for June 2019. Hatched where qa=0.

**Table 3.** Example values for monthly mean tropospheric $NO_2$ column and uncertainties (with percentage of mean tropospheric column) for a few selected locations for January and June 2019 (units: $10^{15}$ molecules $cm^{-2}$). The values for Amsterdam, Beijing, and Rio de Janeiro were taken from one grid cell in the city centre location (52.3N 4.9E for Amsterdam, 39.9N 116.3E for Beijing, and 22.9S 43.1W for Rio de Janeiro). The value for Africa biomass burning represents the location with the highest observed monthly mean value in the sub-equatorial region on the continent of Africa (in a box between latitudes 16S and 16N and longitudes 17.5W and 42E). The uncertainty estimates are with the a priori uncertainty, which does not need to be included when using the averaging kernel.

| | | Amsterdam | Beijing | Rio de Janeiro | Africa biomass burning |
|---|---|---|---|---|---|
| $\bar{x}$ | Jan | 5.45 | 32.28 | 2.87 | 7.20 |
| | Jun | 4.58 | 11.04 | 4.54 | 14.63 |
| $\bar{\sigma}_{total}$ | Jan | 1.15 (21%) | 5.50 (17%) | 0.38 (13%) | 1.03 (14%) |
| | Jun | 0.78 (17%) | 1.72 (16%) | 0.59 (13%) | 2.10 (14%) |

to the averaging of random errors the L3 uncertainty is lower than the uncertainty in L2 orbits, despite the introduction of representativeness errors. The average relative uncertainty in valid L2 pixels in January 2019 in Amsterdam is 52%, compared to the 21% in the L3 dataset. In Beijing the average relative L2 uncertainty is 28%, compared to the 17% in L3 in January 2019.

The separate uncertainty sources are shown in Figure 5 for January 2019. Over unpolluted areas the largest source of uncertainty comes from the estimation of the stratospheric column concentration (Fig. 5b). In polluted regions the largest error

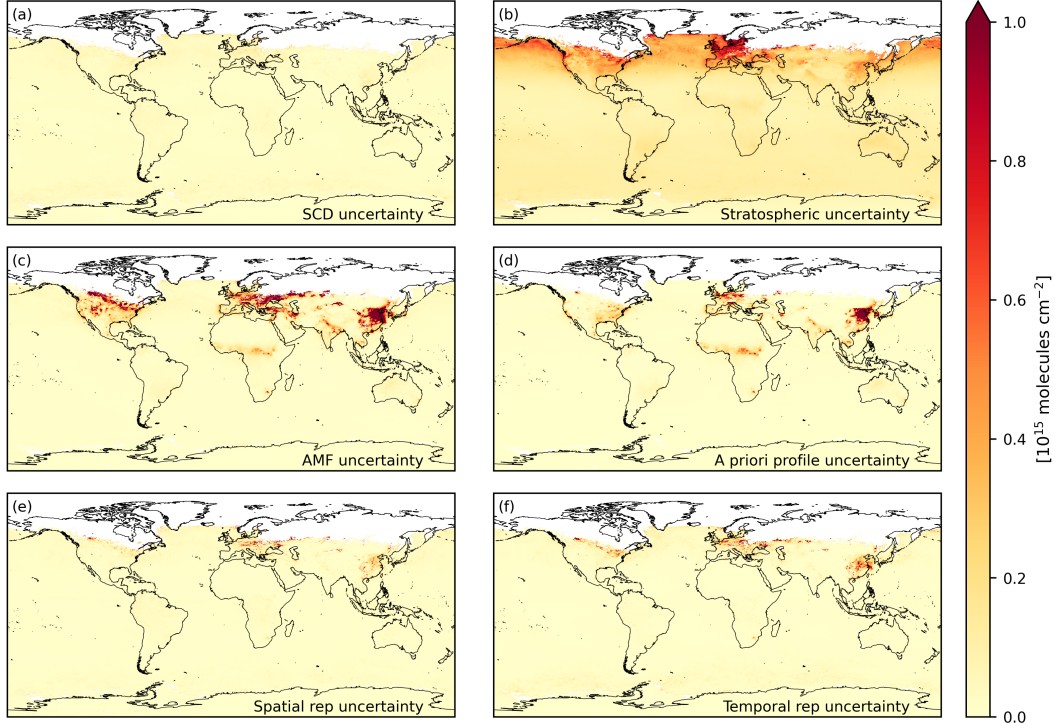

**Figure 5.** All components of tropospheric column density uncertainty using the discussed spatial and temporal correlations for January 2019 (total uncertainty in Figure 4b).

source is the AMF uncertainty (Fig. 5c). Because the slant column density uncertainty is assumed random over both space
and time, the uncertainty averages out over a large number of observations and is very small in the L3 dataset (Fig. 5a), but
it should be noted that the systematic component of the SCD uncertainty is included in the stratospheric uncertainty estimate.
The spatial and temporal representativeness uncertainty is largest in Europe and East Asia, where the standard deviation of the
tropospheric column is largest, but a minor component in the total uncertainty budget.

Time series of the tropospheric $NO_2$ column for four polluted locations are shown in Figure 6. The available time series is too
short to show long-term changes. The time series in Beijing and Amsterdam show features linked to the pandemic lockdowns
(2020-2021) (Bauwens et al., 2020) and indicate an overall reduction of tropospheric $NO_2$ columns over the 5 years, although
Amsterdam displays a large variability and the decrease may not be significant.

The Global Climate Observing System (GCOS) states requirements for observational datasets of the ECVs, including the
precursors for aerosol and ozone variable $NO_2$ tropospheric column (World Meteorological Organization (WMO) et al.,
2022a). We examine the uncertainties in the $NO_2$ tropospheric column against the GCOS required measurement uncertainty
(Figure 7), which is formulated as a threshold, breakthrough, and goal value (World Meteorological Organization (WMO)
et al., 2022b). The threshold requirement, the minimum requirement to be met to ensure that data are useful, requires the rela-

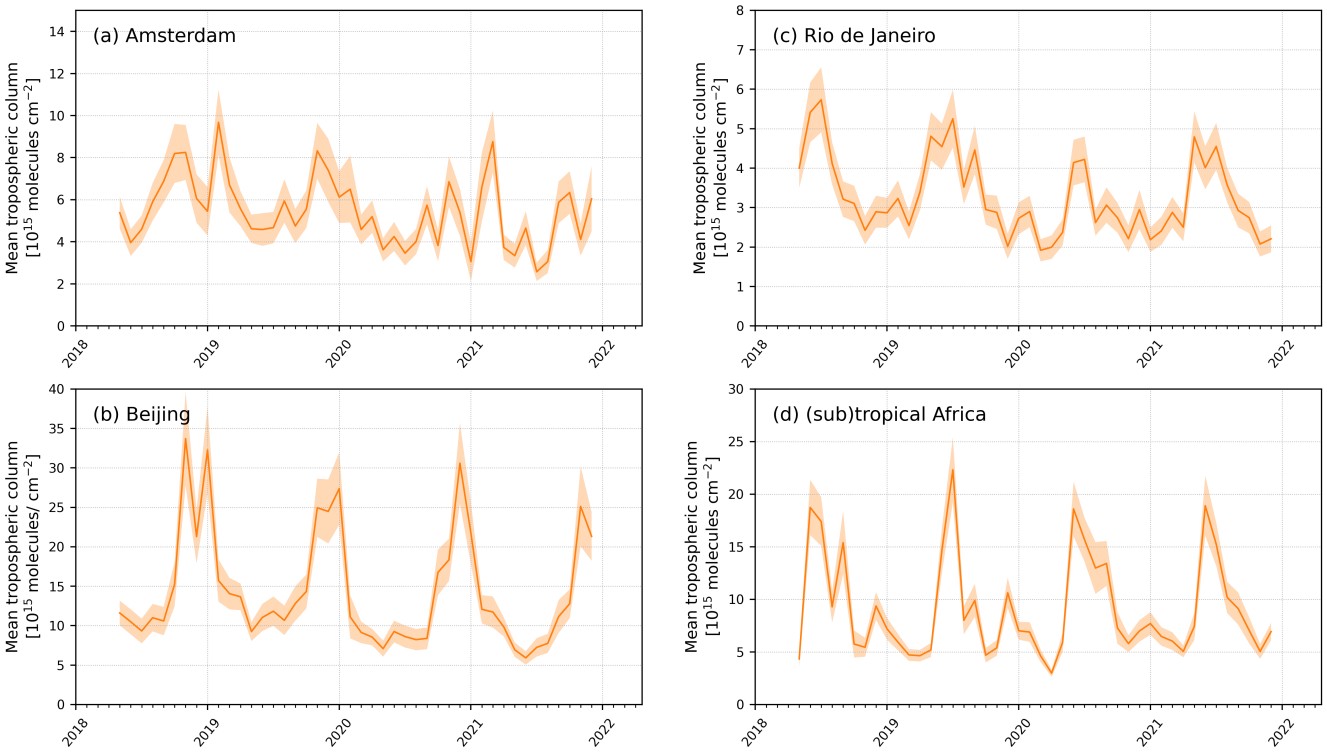

**Figure 6.** Time series of the monthly mean tropospheric column $NO_2$ with the total uncertainty ($\bar{\sigma}_{total}$) for the same locations as in Table 3. The values in the time series represent a single L3 grid cell value and the associated total uncertainty is given as the shaded area.

tive uncertainty to be lower than 100% and the absolute uncertainty to be less than $5 \times 10^{15}$ molecules cm$^{-2}$. The breakthrough requirement represents a significant improvement, and requires the relative uncertainty to be lower than 40% and the absolute

uncertainty to be less than $2 \times 10^{15}$ molecules cm$^{-2}$. Lastly, the goal is reached when the relative uncertainty is lower than 20% and the absolute uncertainty is less than $1 \times 10^{15}$ molecules cm$^{-2}$ (World Meteorological Organization (WMO) et al., 2022b). Due to the averaging of uncorrelated uncertainties in creating the L3 dataset, the GCOS requirements are met more frequently in L3 than in L2 (Figure 7), suggesting L3 data are useful for climate monitoring. In June 2019 over Beijing the absolute uncertainty (and relative uncertainty) drops from an average of $2.81 \times 10^{15}$ molecules cm$^{-2}$ (27.9%) in L2, with most pixels reaching

the threshold requirement, to $1.72 \times 10^{15}$ molecules cm$^{-2}$ (15.5%) in L3, well within the breakthrough requirement (Figure 7c). Over Amsterdam the absolute uncertainty (and relative uncertainty) drops from an average of $1.44 \times 10^{15}$ molecules cm$^{-2}$ (32.9%) in L2, with most pixels reaching the breakthrough requirement, to $0.78 \times 10^{15}$ molecules cm$^{-2}$ (16.9%) in L3, within the goal requirement (Figure 7d).



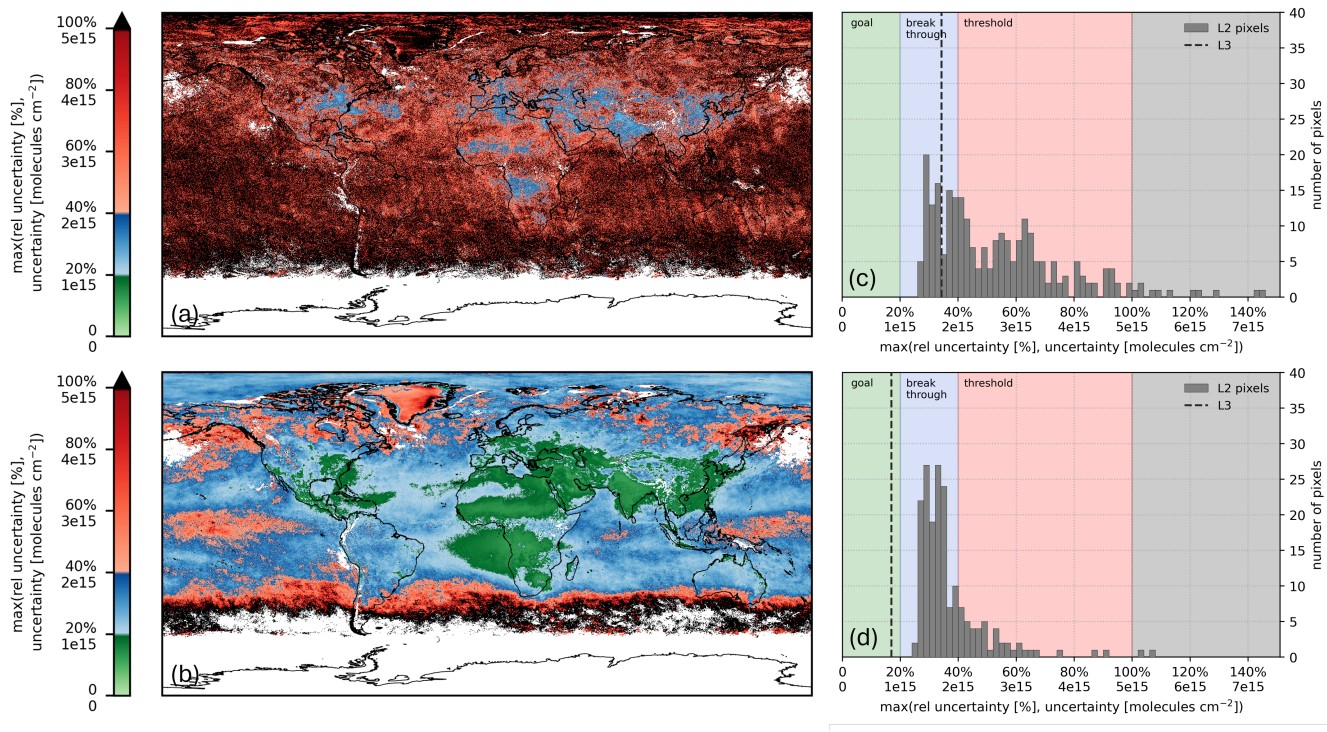

**Figure 7.** June 2019 (a) L2 and (b) L3 dataset as reviewed by the GCOS requirements for measurement uncertainty. Black regions do not fulfil the GCOS requirements, red regions fulfil 'threshold' requirements, blue regions fulfil 'breakthrough' requirements, and green regions fulfil 'goal' requirements. The histograms show the L2 pixel uncertainties that fall in a single $0.2^o\text{x}0.2^o$ L3 grid cell for (c) Beijing and (d) Amsterdam and the grid cell L3 total uncertainty (vertical line).

## 5    Validation

As independent validation, the ESA CCI+ TROPOMI Level 3 $NO_2$ dataset was compared to ground-based remote sensing using the same reference data and methodologies used for the validation of the underlying Level-2 data reported in Verhoelst et al. (2021), in van Geffen et al. (2022a), and in the S5P ATM-MPC quarterly Routine Operations Consolidated Validation Reports (ROCVR, available at https://mpc-vdaf.tropomi.eu/). However, the monthly gridded nature of the L3 data does imply a need for some adaptations in the comparison methodology. The specific aspects and validation results are reported below, per

(sub-)column. The L3 TROPOMI data were all filtered using the $L3\_qa$ value.

### 5.1    Stratospheric-column validation

The L3 TROPOMI stratospheric $NO_2$ columns were compared to the consolidated LATMOS_v3 sunset SAOZ measurements obtained at 10 sites covering mostly clean sites distributed globally from the Southern high latitudes up to the Northern high latitudes (Pommereau and Goutail, 1988). These twilight zenith-sky measurements (Solomon et al., 1987) have an estimated

uncertainty of about 10-14% (Yela et al., 2017; Bognar et al., 2019) and they were adjusted to the average TROPOMI overpass time of the monthly averages (represented in the file as $eff\_frac\_day$) using a model-based photochemical adjustment (Hendrick et al., 2004). For optimal spatial co-location, we compared the SAOZ measurements to the S5P $1.0° \times 1.0°$ L3 grid cell covering the center of the SAOZ observation operator. This procedure accounts for the large horizontal smoothing and offset in the SAOZ measurement sensitivity towards the setting sun (Lambert et al., 1997; Verhoelst et al., 2015). An illustration

of such a comparison, at the Observatoire de Haute Provence in France, is shown in Fig. 8. This comparison shows excellent agreement, within the uncertainties of each product.

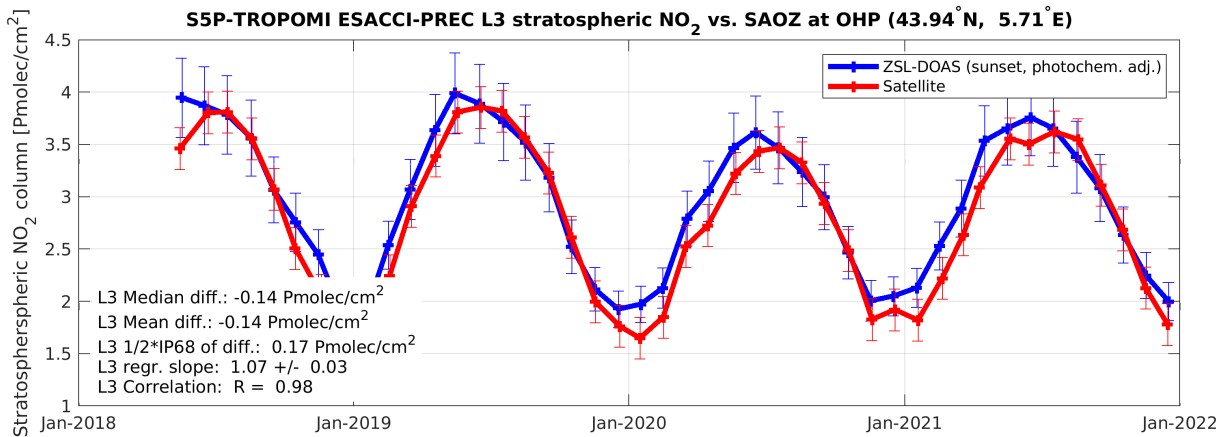

**Figure 8.** Time series of co-located L3 TROPOMI and photochemically-adjusted sunset SAOZ stratospheric $NO_2$ column measurements at the Observatoire de Haute Provence (France).

The network-wide results are summarized in Fig. 9. Overall, these stratospheric column comparisons yield results very similar to those for the underlying L2 (Verhoelst et al., 2021), with a virtually insignificant network-mean bias and a dispersion of typically around $0.2 \times 10^{15}$ molecules $cm^{-2}$. The somewhat larger bias and dispersion over Paris is probably related to

tropospheric contamination in the SAOZ measurements.

## 5.2   Tropospheric column

The L3 TROPOMI tropospheric $NO_2$ columns were compared to the MAX-DOAS tropospheric column data (Hönninger and Platt, 2002) collected from various sources and harmonized (in terms of file format) through the NIDFORVAL project for the operational S5P L2 validation. Total uncertainty estimates on these tropospheric VCD measurements are of the order of

7%–17% in polluted conditions, including both random (around 3% to 10%, depending on the instrument) and systematic (11% to 14%) contributions (e.g., Hendrick et al., 2014). MAX-DOAS data obtained within 30 minutes of the underlying L2 S5P data were compared to $0.2° \times 0.2°$ L3 grid cells covering the station location. To ensure good temporal representativeness, only those sites at which at least a full year of comparisons could be made, were retained. This yielded 8 sites, covering moderately to severely polluted conditions. The most polluted case (e.g., Pinardi et al., 2020), Xianghe in China, is analysed in Fig. 10,

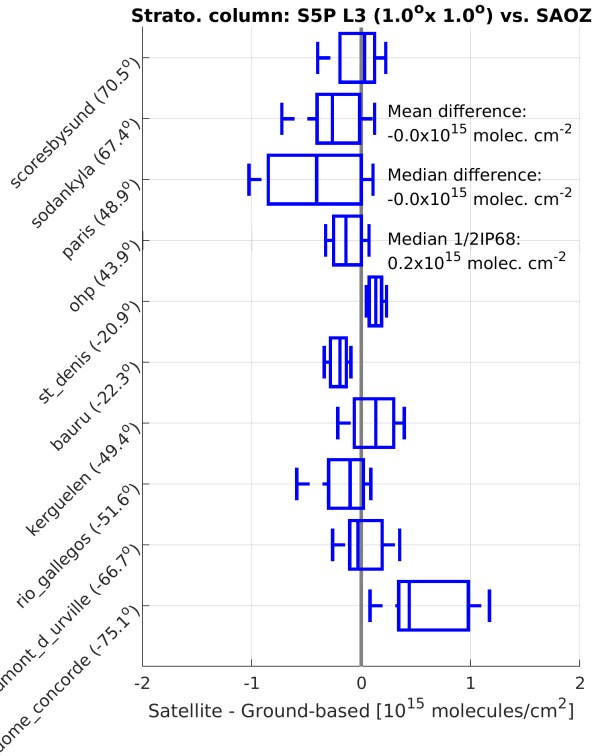

**Figure 9.** Box-and-whisker plots summarizing from pole to pole the bias and spread of the difference between the L3 TROPOMI and ground-based SAOZ data. The median difference is represented by a vertical solid line inside the box that marks the 25% and 75% quantiles. The whiskers cover the 9%–91% range of the differences. Values between brackets in the labels denote the latitude of the station.

revealing a slightly better correlation and smaller dispersion of the differences than observed in the validation of the underlying L2 data. This is most likely a due to the reduced measurement and atmospheric noise in the monthly averages. The mean and median difference are however more negatively biased, which canb be attributed the poorer spatial resolution of the L3 data set compared to the constituent L2 data.

     The network-wide results for the MAX-DOAS comparisons are visualized in Fig. 11. These confirm the relatively small

dispersion of the differences, typically about 12%, which is well within the combined prognostic uncertainty budget (quadratic sum of 20% uncertainty on the TROPOMI L3 data and probably at least 10% on the monthly averaged MAX-DOAS data, depending on how one propagates the systematic and random components of the MAX-DOAS uncertainty). Also confirmed is the strong negative bias in TROPOMI $NO_2$, to be understood as a combination of the L2 negative bias and additional systematic differences related to the spatial smearing in the L3 data. Unexpected is the more pronounced negative bias at the relatively

clean sites, which is opposite to the behaviour observed for the underlying L2 data. This result is not confirmed by the total column comparisons described in Sect. 5.3 and may be a case of small-number statistics or peculiarities at these individual sites.



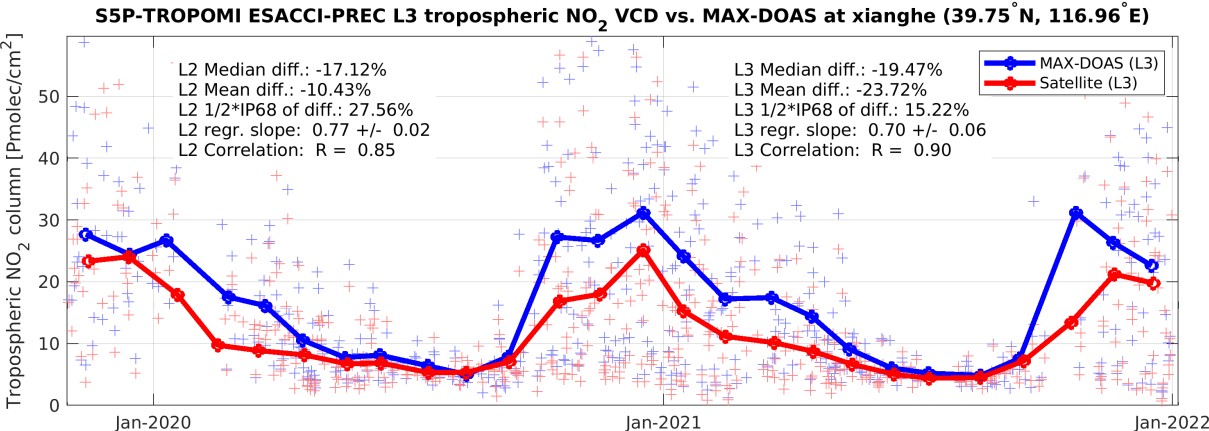

**Figure 10.** Time series of co-located TROPOMI and MAX-DOAS tropospheric $NO_2$ column measurements at Xianghe (China). Light markers represent the co-located L2 data from the current operational processor (v2.4 RPRO, as used for ROCVR #24), the solid lines represent the L3 data.

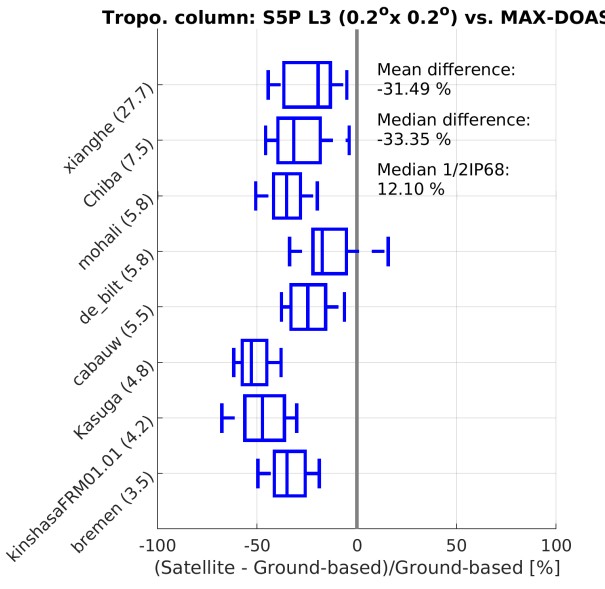

**Figure 11.** Similar to Fig. 9 but representing the agreement between the S5P-TROPOMI and MAX-DOAS tropospheric $NO_2$ column measurements, limited to those sites for which a full year of comparisons is available. Sites are ordered by mean tropospheric $NO_2$ VCD, cleaner sites at the bottom, more polluted sites at the top.

## 5.3 Total-column validation

The TROPOMI L3 total $NO_2$ columns, calculated as the sum of the tropospheric and stratospheric columns provided in
the data files, were compared to Pandora direct-sun measurements (v1.8) from the Pandonia Global Network (PGN, https:





//pandonia-global-network.org/). These measurements have a random error uncertainty of about $0.27 \times 10^{15}$ molecules cm$^{-2}$ and a systematic error uncertainty of $2.7 \times 10^{15}$ molecules cm$^{-2}$ (Herman et al., 2009). All Pandora measurements satisfying the recommended PGN quality filtering and obtained within 30 minutes from the satellite effective fractional day ($eff\_frac\_day$ in the data files) were averaged and only the $0.2° \times 0.2°$ L3 grid cell covering the instrument location was used. To ensure good

temporal representativeness, only those sites at which at least a full year of comparisons could be made, were retained. This yielded 8 sites, covering both unpolluted rural conditions and severe pollution, e.g., in the megacity of Mexico City. Figure 12 demonstrates the agreement at this most polluted site for both the L3 product and the current operational L2 product, which is the v2.4 full mission reprocessed (RPRO) data set for this period of S5P measurements.

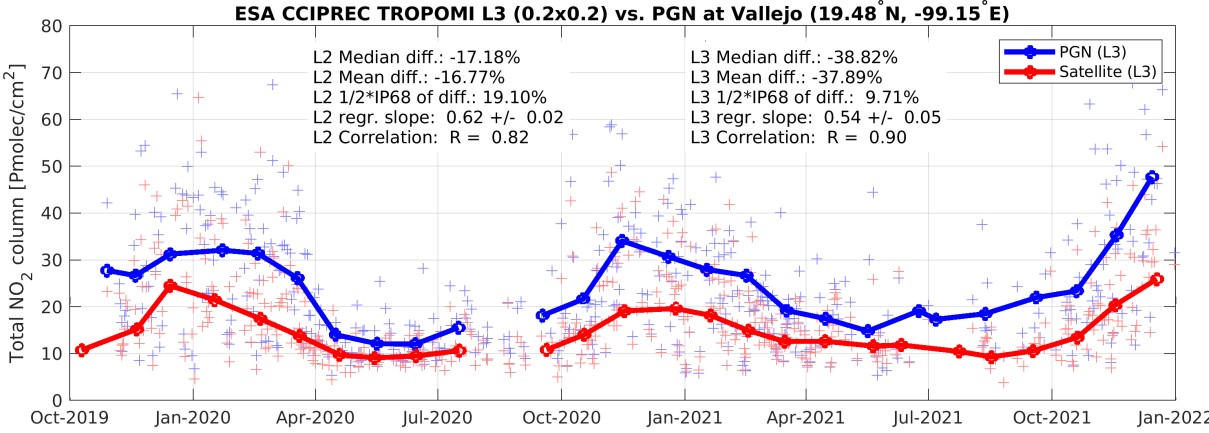

**Figure 12.** Time series of co-located TROPOMI and PGN total NO$_2$ column measurements at Vallejo (Mexico City). Light markers represent the co-located L2 data from the current operational processor (as available on the ATM-MPC VDAF AVS), the solid lines represent the L3 data.

As for the tropospheric VCD comparisons, the correlation between TROPOMI L3 and Pandora data is excellent ($r \sim 0.9$)

for these comparisons in a highly polluted environment, and even better than at L2. Again, as for the tropospheric column comparisons, the NO$_2$ underestimation already seen in L2 is more pronounced for the L3 NO$_2$ product with a median diff of about -35% (versus -17% for L2) and a regression slope of 0.54 (versus 0.62 for L2). For a pollution hot spot such as Mexico city, this is most likely due to the coarse L3 product resolution ($0.2°$), which tends to spatially smear out gradients compared to the pixel-resolution L2 data ($0.05°$). On the other hand, thanks to the temporal averaging, the dispersion of the

differences between L3 and PANDORA NO$_2$ is significantly reduced, from more than 15% down to about 10%. The network-wide results are summarized in Fig. 13. The tendency that TROPOMI NO$_2$ show a more pronounced underestimation for larger total NO$_2$ column values is in line with the L2 validation results which show virtually no underestimation (or even a very slight overestimation) at the cleanest sites, where the NO$_2$ column is dominated by the stratospheric contribution, and an underestimation up to 20-30% at the most polluted sites (Verhoelst et al., 2021, and updates in the ROCVR). These total

column results therefore do not confirm the more pronounced negative biases observed in the MAX-DOAS comparisons at sites with only moderate pollution (when compared to more polluted sites).



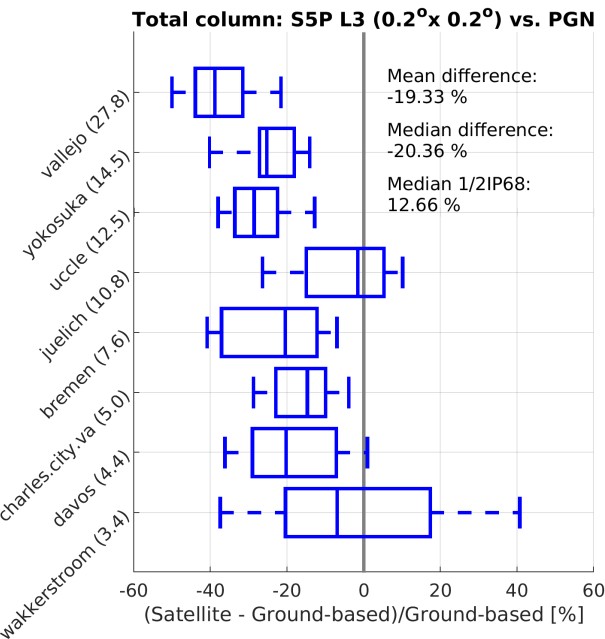

**Figure 13.** Similar to Fig. 9 but representing the agreement between the S5P-TROPOMI and PGN total $NO_2$ column measurements, limited to those sites for which a full year of comparisons is available. Sites are ordered by mean total $NO_2$ VCD, cleaner sites at the bottom, more polluted sites at the top.

## 6  Conclusions

We developed a comprehensive level-3 (L3) dataset of gridded and averaged tropospheric $NO_2$ column retrievals (v2.3.1) from the TROPOMI sensor, covering the period from May 2018 to December 2021. The dataset is available at multiple spatial resolutions ($0.2° \times 0.2°$, $0.5° \times 0.5°$, and $1° \times 1°$) on a monthly timescale. Using a tiling (superobservation) approach, we ensured that (1) valid $NO_2$ retrievals are gridded according to pixel area, and (2) temporal averages are calculated as weighted means, with weights reflecting the representativeness of superobservations for specific days. This L3 dataset is applicable for clear-sky or low-cloud conditions at TROPOMI's overpass time of 13:30.

Realistic uncertainties were derived by propagating L2 retrieval uncertainties. To address spatial and temporal error correlations, we considered errors in the stratosphere-troposphere separation and air mass factor calculations, which are partly spatially correlated. Representativeness uncertainty, stemming from incomplete coverage of grid cells by valid L2 retrievals (e.g., due to cloud cover), was found to be more significant in polluted regions. The combined L3 uncertainty accounts for both measurement uncertainties (including spatial representativeness errors) and temporal representativeness uncertainties. Analysis showed that 30% of retrieval uncertainties persist over a month due to error correlations in the stratosphere-troposphere separation and air mass factor calculations.





Over polluted areas, the L3 dataset showed reduced uncertainties compared to averaged L2 retrievals, demonstrating the effectiveness of averaging large numbers of observations. For example, while monthly average L2 uncertainties are 30–50% over Beijing and Amsterdam, they drop to 20% or less in the L3 dataset, meeting the GCOS 'breakthrough' and even 'goal' requirements. Validation against ground-based PANDORA measurements revealed a consistent temporal correlation but with a low bias of 20%, partly attributed to more pronounced spatial smearing in the L3 product.

This monthly mean L3 TROPOMI tropospheric $NO_2$ dataset offers a coherent and much-reduced (in size) data record, making it suitable for atmospheric chemistry studies, for evaluating atmospheric models and analyzing spatiotemporal $NO_2$ trends. The methods presented here can be replicated when creating L3 datasets for other atmospheric gases and other Earth Observation L3 datasets. Future work will include applying the presented methods on $NO_2$ retrievals from OMI and combine TROPOMI and OMI observations to create a decades-long consistent CDR.

## 7   Code and data availability

Data described in this manuscript can be accessed at repository under data doi https://doi.org/10.21944/CCI-NO2-TROPOMI-L3. (KNMI, 2025). The generated L3 dataset will be made available on the ESA Climate Change Initiative Open Data portal. The software to create the spatial average is available at https://doi.org/10.5281/zenodo.10726644. Software to generate the temporal-spatial mean resulting in the L3 dataset is available at https://doi.org/10.5281/zenodo.14505524. The TROPOMI L2 v2.3.1 $NO_2$ dataset which is the input for the generated dataset is not the operational version and thus no longer publicly available. A sample of the data can be shared upon request. TROPOMI $NO_2$ L2 v2.4 is available on the Copernicus Data Space Ecosystem (https://doi.org/10.5270/S5P-9bnp8q8). The reference data used for the ground-based validation are available from the NDACC Data Host Facility at www.ndacc.org (SAOZ and selected MAX-DOAS data) and from both the PGN website (https://www.pandonia-global-network.org/) and ESA's Validation Data Center (EVDC, https://evdc.esa.int/) for the Pandora data.

## Appendix A:  L2 observations in the descending node

During polar summer, the polar regions are marked by 24 hours daylight. In the Northern Hemisphere, this means that the polar region on the 'backside' of the Earth is experiencing daylight from May to August, and observations are made in the descending node of the satellite. These observations in the descending node are included in the L2 $NO_2$ product and not quality-flagged in the qa-value before version 2.7.

However, we find that the observations in the descending node of the orbit show structurally different retrieval results for the tropospheric vertical $NO_2$ column in the Arctic region (north of 60° latitude) than the observations from the ascending node in June (Figure A1a). While retrievals in the ascending part of the orbit show a median value of $0.2 \times 10$ molecules cm$^{-2}$, retrievals in the descending part show a long tail of negative tropospheric $NO_2$ values and the mode of the probability density





curve is negative. This difference between results from the ascending and descending part of the orbits is less obvious in the Antarctic region in December (Figure A1b).

A TROPOMI validation report (Lambert et al., 2023) has shown that TROPOMI underestimates tropospheric $NO_2$ columns compared to ground-based Pandora instruments at the high-latitude locations of Ny-Ålesund and Eureka by about 15%. The

discrepancy between tropospheric $NO_2$ column observations from the descending node and ascending node could play a role in this underestimation.

The discrepancy between retrievals done in the descending and ascending node is caused because the descending node observations are not included in the data assimilation in TM5-MP and thus the stratospheric columns are more uncertain and often overestimated (Figure A1c). Based on these results, we recommend to not use retrievals from the descending part of the

orbit. From TROPOMI L2 version 2.7 onwards, the qa-value is adjusted to include flags for the descending orbital observations.



**Figure A1.** NO$_2$ vertical column observations (qa>0.75) in ascending and descending mode for the troposphere (a) and stratosphere (c) north of 60° latitude from orbits in June 2019 and for the troposphere (b) and stratosphere (d) south of -60° latitude from orbits in December 2019.





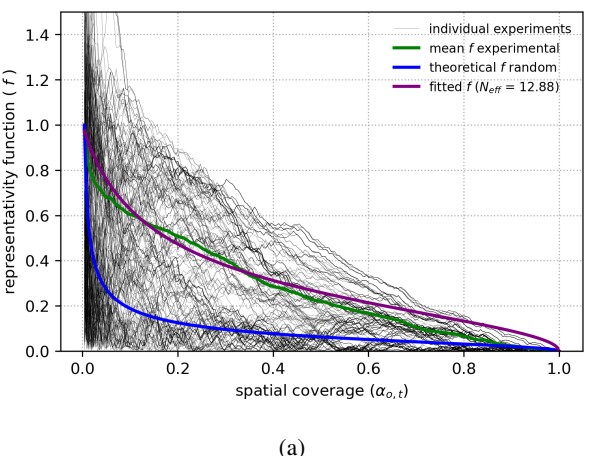
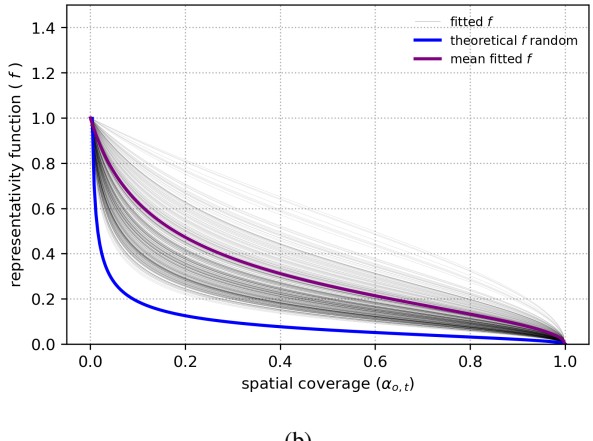

(a)                                (b)

**Figure B1.** Results of repeatedly sampling grid cells in a systematic way to calculate the representativeness function $f$. (a) One single grid cell (polluted, $x_{o,t} \geq 1.8 \times 10^{15}$ molecules cm$^{-2}$) sampled 200 times. The grey lines represent 200 individual experiments sampling the same superobservation grid cell. The green line is the mean of the individual experiments and the purple curve is the representativeness function in equation 6 fitted to the green line. The blue curve shows the theoretical representativeness function in the case of random sampling (result if grid cell is not sensitive to systematic sampling). (b) Results of sampling 100 grid cells in polluted regions. The grey lines is a collection of purple lines from (a), the purple curve is the mean of the grey curves. The blue line again shows the theoretical random solution.

## Appendix B: The number of effective observations

The number of effective observations $N_{eff}$ characterizes how sensitive a region is to systematic sampling, which can occur due to for example a cloud field covering part of a grid cell. The lower the value for $N_{eff}$, the more sensitive a region is to systematic sampling. In a region where the uncertainty in NO$_2$ columns is dominated by noise, usually a relatively unpolluted region, the

$N_{eff}$ is high. In such a region, the draw of one retrieval can be enough to characterize the spatial ensemble on a whole. In an area with strong pollution gradients, usually a polliuted area, multiple draws are needed for proper characterization. For example, a grid cell covering a city, effectively consists of two types of NO$_2$ observations: those over the polluted city and those over the unpolluted nearby rural area, and such areas are thus sensitive to systematic sampling.

    The value for $N_{eff}$ was estimated by Rijsdijk et al. (2024) by repeatedly sampling a fully-covered grid cell in a systematic

way (starting with one random pixel and then adding neighbouring pixels) and calculating the representativeness (the absolute difference between the mean of observed pixels and the true mean of the fully covered grid cell when including all pixels). The representativeness function $f$ in equation 6 was then fitted to the mean of these experiments, providing the number of effective observations $N_{eff}$ for this superobservation grid cell (Figure B1a). For this polluted grid cell, with resolution 0.2$^{\text{o}}$ × 0.2$^{\text{o}}$ and TROPOMI pixels of 5.5 × 3.5 km$^2$, the number of effective observations $N_{eff}$ is somewhat smaller than 13.

This method was repeated for multiple superobservation grid cells. The results showed that polluted regions ($x_{o,t} \geq 1.8$ molecules cm$^{-2}$) have a lower value for $N_{eff}$ than unpolluted regions ($x_{o,t} < 1.8$ molecules cm$^{-2}$), making it more sensitive





**Table B1.** The ratio $N/N_{eff}$ for TROPOMI $NO_2$ in unpolluted and polluted ($x_{o,t} > 1.8 \times 10^{15}$ molecules cm$^{-2}$) regions for different spatial resolutions of the superobservation grid. The ratio increases with superobservation area (degree$^2$).

|  | $N/N_{eff}$ unpolluted areas | $N/N_{eff}$ polluted areas |
|---|---|---|
| 0.2°x0.2° | 1.376 | 3.933 |
| 0.5°x0.5° | 1.890 | 7.392 |
| 1.0°x1.0° | 3.724 | 19.746 |
| 2.0°x2.5° | 13.508 | 85.634 |

to systematic sampling (Rijsdijk et al., 2024). The ratio $N/N_{eff}$, where $N$ is the number of pixels (valid or not) in a super-observation, was determined for $NO_2$ retrievals from TROPOMI in Rijsdijk et al. (2024) to be linearly dependent on the grid resolution. This linear relationship was determined for both polluted and unpolluted superobservations by Rijsdijk et al. (2024).

For a superobservation resolution of 0.2°x0.2° this results in $N/N_{eff} = 1.376$ for unpolluted areas which are not sensitive to systematic sampling and $N/N_{eff} = 3.933$ for polluted areas which are more sensitive to systematic sampling (Table B1). These ratios are used to determine the number of effective observations $N_{eff}$ for each grid cell in the superobservation. When the $N_{eff}$ is equal to the number of pixels $N$, and thus the ratio $N/N_{eff}$ equals 1, systematic sampling has no effect. The ratios $N/N_{eff}$ would need to be redetermined for another substance or sensor (with a different pixel resolution) following the methods in

Rijsdijk et al. (2024).



## Appendix C: Contents of L3 dataset

**Table C1.** Overview of the variables, units and types in the main L3 data output product file.

| name/data | symbol | unit | data set name |
|---|---|---|---|
| air-mass factor | $M^{\text{trop}}$ | 1 | tropospheric_NO2_column_number_density_amf |
| | $M$ | 1 | total_NO2_column_number_density_amf |
| averaging kernel | $\mathbf{A}$ | 1 | NO2_averaging_kernel |
| cloud radiance fraction | $w_{\text{NO2}}$ | 1 | cloud_fraction |
| cloud pressure | $p_{\text{c}}$ | hPa | cloud_pressure |
| grid cell coordinates | $\delta_{\text{geo}}$ | ° | latitude |
| | $\vartheta_{\text{geo}}$ | ° | longitude |
| grid cell corners | $\delta_{\text{geo}}$ | ° | latitude_bounds |
| | $\vartheta_{\text{geo}}$ | ° | longitude_bounds |
| land/water classification | - | 1 | land_water_mask |
| number of pixels used | - | 1 | no_observations |
| in averaging | - | 1 | tropospheric_NO2_column_number_density_count |
| profile layers | $N_l$ | 1 | layer |
| quality assurance value | - | 1 | qa_L3 |
| slant column density | $N_{\text{s,NO}_2}$ | molecules cm$^{-2}$ | NO2_slant_column_number_density |
| | $N_{\text{s,NO}_2}^{\text{trop}}$ | molecules cm$^{-2}$ | NO2_slant_column_number_density_troposphere |
| | $\sigma_{N_{\text{s,NO}_2}}$ | molecules cm$^{-2}$ | NO2_slant_column_density_uncertainty |
| surface albedo | $A_{\text{s}}$ | 1 | surface_albedo |
| surface pressure | $p_{\text{s}}$ | hPa | surface_pressure |
| time | - | days | eff_date |
| | - | 1 | eff_frac_day |
| | $t$ | date | time |
| TM5 pressure level | $A_l^{\text{TM5}}$ | hPa | tm5_sigma_a |
| coefficients | $B_l^{\text{TM5}}$ | 1 | tm5_sigma_b |
| vertical column density | $N_{\text{v,NO}_2}^{\text{trop}}$ | molecules cm$^{-2}$ | tropospheric_NO2_column_number_density |
| | $N_{\text{v,NO}_2}^{\text{strat}}$ | molecules cm$^{-2}$ | stratospheric_NO2_column_number_density |
| vertical column uncertainty | $\bar{\sigma}$ | molecules cm$^{-2}$ | tropospheric_NO2_column_number_density_temporal_std |
| | $\bar{\sigma}_{\text{total}}$ | molecules cm$^{-2}$ | tropospheric_NO2_column_number_density_total_uncertainty |
| | $\bar{\sigma}_{\text{total,kernel}}$ | molecules cm$^{-2}$ | tropospheric_NO2_column_number_density_total_uncertainty_kernel |





*Author contributions.* I.A.G. developed the methodology, created software, and carried out the analysis and writing of the initial draft. K.F.B. co-developed the methodology, contributed to the conceptualization and writing of the manuscript. I.A. and and H.J.E. contributed to the conceptualization and analysis of results. P.R. developed the methodology and software for the spatial averaging procedure. T.V., S.C., G.P., and J.-C.L. and M.V.R. were involved in the validation of the dataset. All authors contributed to review and editing of the manuscript.

*Competing interests.* The authors declare that they have no conflict of interest.

*Acknowledgements.* This work was carried out as part of the ESA CCI+ Precursors for Aerosols and Ozone project, funding the contributions from I.A.G., I.A., K.F.B., H.J.E., T.V., S.C., G.P., J.-C.L., and M.V.R..

Sentinel-5 Precursor is a European Space Agency (ESA) mission on behalf of the European Commission (EC). The TROPOMI payload is a joint development by ESA and the Netherlands Space Office (NSO). The Sentinel-5 Precursor ground segment development has been funded by ESA and with national contributions from the Netherlands, Germany, and Belgium. This work contains Sentinel-5P TROPOMI data, processed in the operational framework or locally at KNMI.





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
