# Peer review of "TROPOMI Level 3 tropospheric NO2 Dataset with Advanced Uncertainty Analysis from the ESA CCI+ ECV Precursor Project"

_Earth System Science Data, 2024_

## Author Comment (AC1)

Reply to "RC2: 'Comment on essd-2024-616', Anonymous Referee #2, 14 Apr 2025", Citation: https://doi.org/10.5194/essd-2024-616-RC2

**Glissenaar et al.**

This manuscript presents a global monthly (Level 3) data set of tropospheric nitrogen dioxide, specifically as a climate data record for the ESA Climate Change Initiative project. This product is accompanied by uncertainties in the Level 3 data set. Careful attention is taken to discuss how random uncertainties cancel when averaging over both space and time. The product is evaluated against several datasets.

Overall, the manuscript is written clearly with fairly comprehensive detail and insightful figures that support the text. I thought the derivation and discussion of uncertainties as the observations are averaged over space and then time was extremely valuable and comprehensive. While important enough to accompany this dataset anyway, the topic is also very useful to the community more widely. Quantifying how the average relative uncertainty in valid L2 pixels compares with the uncertainty in the L3 monthly product is very enlightening. I almost wished there were more examples in Table 3 and discussed in Lines 389-393, but I understand that a user could simply consult the data itself.

I thought the product evaluation was robust and comprehensive. Sources of discrepancies were discussed.

**We'd like to thank the reviewer for their thoughtful observations and recognition of the relevance of this study.**

The first anonymous reviewer comment points to the work by Rijsdijk et al. (2024), which may not get enough attention in the motivation of this manuscript (throughout the introduction, for example), and I agree with this assessment. If I understand correctly, the main contribution of this manuscript from an algorithmic perspective (compared to Rijsdijk et al., 2024) is the addition of temporal averaging in the Level 3 product here. Perhaps the authors could spend a bit more time really clarifying exactly which other developments (if applicable) are introduced here.

We agree with this assessment. We have introduced a paragraph about Rijsdijk et al.'s work in the introduction. We have also included another reference to Rijsdijk et al. (2025) in the overview of methods in Line 130. We have also included a note in

**Line 224 (the start of the temporal averaging section) that this is new in the present study.**

I also had a question about the potential cloud selection bias that could be introduced in the averages, *in addition to* the spatial representativeness uncertainty. I did not see that addressed here in this text. I see two points that are worth considering:

(1) Using this Level 3 dataset as a "clear sky average": for example, comparing with *coincidently sampled cloud-free model data*, the importance of a potential bias in your monthly average from selecting relatively cloud-free conditions seems to be avoided. The two quantities (cloud free satellite average and cloud-free model average) are directly comparable and the uncertainty estimate reported in the Level 3 product stands as appropriate for this application.

**Yes, this is correct. We have now included a note in the dataset description (section 4.1) clarifying this.**

(2) Other users may expect to consider the uncertainty in this Level 3 product as an uncertainty in the *true monthly mean*. In this case, it seems as though the reported uncertainty doesn't actually account for a potential bias due to selection of cloud-free conditions. Cloudiness could adjust photolysis rates and chemical lifetimes, or it could be correlated with atmospheric transport conditions that may bring cleaner or more polluted conditions to a particular location. The potential "bias" in a monthly average due to selecting cloud-free conditions could conceivably be positive *or* negative depending on location. Reviewer #1 alludes to this in their first comment on Section 3.3.1: "If I understood correctly, superobservations, which are probably clear-sky observations, have a higher weight; do these have a tendency to lower/higher NO2 concentrations and create a bias?". I suspect the answer could be either, and depends on local chemistry and atmospheric transport conditions relative to the location of pollutant hotspots (e.g., Geddes et al., 2012: https://doi.org/10.1016/j.rse.2012.05.008).

We have added a discussion on the difference between the clear-sky and true monthly average in the dataset description (section 4.1), including what the assessed uncertainty represents (not the uncertainty to the true average, but indeed to the clear-sky average) and how overcast conditions could affect the true monthly mean. We studied this issue before in Boersma et al. (2016), who compared the true NO2 monthly mean to the clear-sky NO2 monthly mean and found that the former is biased high by up to 25% over polluted regions. While not exactly within the scope of this study (which focuses on propagating errors appropriately across spatial and temporal averaging), I think this point deserves some brief discussion/clarification, and especially advice to users when considering the data as a monthly average (and not when just comparing it to other clear-sky sampled data sets).

**We agree, and adding the discussion and reference to the Boersma et al. (2016)-study as outlined above will help to that effect.**

Finally, I wondered about the cadence of updates for this particular product. The data in this manuscript is for May 2018 to December 2021. Can users expect this Level 3 algorithm and data product to be regularly updated, and if so, at what frequency? Will the record soon be available for more recent TROPOMI years? What are the plans for long-term processing?

The L2 v2.3.1 (consistent with OMI L2) which is used in this dataset is only available until December 2021, and will not be extended. Therefore we cannot extend this specific L3 dataset.

We are planning to process TROPOMI 2.4/2.5 (consistent algorithm version) for May 2018 – Nov 2023 into a L3 dataset using the same methods as discussed in this paper.

TROPOMI L2 will be reprocessed (2.9) in early 2026. This could give an opportunity to have a L3 dataset over a long consistent period.

---

## Author Comment (AC2)

**Reply to "RC1: 'Comment on essd-2024-616', Anonymous Referee #1, 30 Mar 2025", Citation**: https://doi.org/10.5194/essd-2024-616-RC1

*Glissenaar et al.*

The paper by Glissenar et al. presents the creation of a monthly global Level 3 dataset of TROPOMI tropospheric NO2 VCDs and uncertainties at spatial resolutions of 0.2°, 0.5°, and 1°. The spatiotemporal averaging of the NO2 data and its uncertainty is described in detail. Spatial and temporal error correlations for all sources of uncertainty in the L2 retrieval are analyzed. The total relative uncertainty in the resulting Level 3 dataset is analyzed globally for different levels of pollution. The tropospheric, stratospheric, and total vertical NO2 column is validated with ground-based measurements.

The study covers the important topic of creating L3 datasets with proper uncertainty analysis, which have received less attention from the scientific community than the L2 data product. This L3 dataset is of interest for atmospheric chemistry studies, for evaluating atmospheric models and analyzing spatiotemporal NO2 trends. The study contains important analyses of the created L3 dataset. However, in some parts the paper is hard to follow and contains some inconsistencies which could be improved by doing some major revisions and addressing the comments raised below.

We thank the reviewer for recognizing the relevance of our study.

**General comments:**

The study by Rijsdijk et al (2024) is an important study for your analysis. I think it would be necessary to introduce the main study results in the introduction before you reference it several times throughout your study.

We agree that there could be more emphasis on Rijsdijk et al (2024) in the introduction already, and not wait for the methods section. We will do this in the revised version.

Chapter 3 Methodology: This chapter describes how you have averaged the L2 NO2 column data and handled the uncertainties. It is the main part of your study, however, I think it is hard to follow, and would benefit from a detailed check: Does the reader know this variable already? Mention that, e.g., uncertainty x is described

in more detail in subsection x. Please, check for consistency in variables and their indices. See also specific comments below.

We have done a check if all variables have been introduced properly. With your technical comments we noticed we wrongly introduced a new variable $g$, which has now been changed into $f$ which was introduced before. We have also changed the $s$ that stood for measurement source in $\sigma_{m,s}$ into $c$ for measurement contribution in order to avoid confusion with $\sigma_{rs}$ for spatial representativity uncertainty. We believe all variables have been introduced and are consistent, although we do register there are many.

Section 3.3.1: Isn't this weighting creating a bias? If I understood correctly, superobservations, which are probably clear-sky observations, have a higher weight; do these have a tendency to lower/higher NO2 concentrations and create a bias? Can this be neglected in your averages, and using only observations with cloud radiance fraction < 0.5?

Superobservations $x_{o,t}$ are based on valid clear-sky (cloud radiance fraction < 0.5) observations already. The weighting in Eq. (8) takes into account that different superobservations have different degrees of representativeness, driven by the number of valid L2 pixels used for calculating the superobservation. If we would not weigh the superobservations according to their uncertainties, we would skew the L3 columns to the least representative superobservations, which is something to avoid.

The satellite observations are not able to determine the full tropospheric column in overcast situations. Therefore, the L3 product will be a clear-sky NO2 product by definition. We clarify in the dataset description (4.1) that the dataset represents clear-sky conditions.

Inconsistent use of abbreviations SCD and $N_s$ for the slant column density, same for the air mass factor with AMF and $M^{tr}$.

We have removed mentions of SCD and VCD and instead wrote the full form (except for when clarification was necessary in a figure caption). We have kept the use of both AMF and $M_{tr}$ and use the former in the text and the latter as a mathematical symbol when showing the equations (to clarify it is the tropospheric air mass factor).

The word column is often used without defining whether it is slant or vertical.

We will check this. Perhaps in 3.1, the use of starting point does not work and we should spell out that this concerns tropospheric columns everywhere.

**Specific comments:**

Line 8: Spatial error correlations arise not only from these two, but also from the a priori model. Add "mainly" or mention also the a priori model.

Agreed. We will add "mainly".

Line 9: Is it important here that the albedo climatology has a coarse grid, because it is coarse compared to the TROPOMI pixel but similar to the grid of the L3 dataset?

The fact that the albedo climatology is coarse relative to TROPOMI pixel scale is important, because it implies that when the albedo is biased, it will impact the albedo from one pixel to the next, in other words incur spatially correlated errors.

Line 10: You name the temporal error correlation to be 30%. Before you mention the spatial error correlation, but you are not mentioning a value.

The spatial error correlation from Rijsdijk et al. (2025) (i.e. in the spatial averages) does not have a single value. In Rijsdijk et al. (2025) the different contributions to the error in the tropospheric column were assessed separately and for the AMF uncertainty depends on the grid size. As this is not quantifiable in a single sentence, and not a result of this study but a recap of previous studies, we have not included this in the abstract. A full discussion of the spatial error correlations is available in section 3.2.2.

We will include a clarification in the abstract that these spatial error correlations are a result of previous studies. ESSD submission rules state that reference citations should not be included in the abstract.

Line 29: This sentence is incomplete, what kind of compounds do you mean?

We replaced 'compounds' by nitrogen oxides to better express why NO2 is relevant to monitor.

Line 35: I think you should not only mention in-situ but also remote-sensing here.

Remote sensing is mentioned in the next sentence, to emphasize its strength in global coverage.

Line 38: I think it might be a good point to introduce some satellite instruments here, especially the ones like OMI and TROPOMI you mention later, because later it might be good to know, and maybe not obvious for everyone, that OMI is the "precursor" of TROPOMI. You mention OMI in line 83 without introducing it.

Thanks for the suggestion. We now add at the end of the paragraph: *Among the key satellite instruments used to monitor tropospheric $NO_2$ columns are OMI (Ozone Monitoring Instrument) on NASA's Aura satellite and TROPOMI (Tropospheric Monitoring Instrument) on ESA's Sentinel-5 Precursor, which provide long-term, high-resolution datasets critical for understanding air pollution trends.*

Line 74: I would avoid the word observation here because the NO2 is not the observation but the resulting product. You could also mention that there are also other products from TROPOMI, besides NO2.

We now changed the sentence to express that TROPOMI provides tropospheric column data rather than observations: *The TROPOspheric Monitoring Instrument (TROPOMI) (Veefkind et al., 2012) provides data on tropospheric NO2 (and many other trace gases) columns.*

Line 90: This sentence is not clear: Replace "this version" with the version you mean (v2.3.1 ?) Was the qa_value bug corrected in this study? If yes, add this information and what was done. Replace (qa) value with qa_value, which was already introduced.

The qa-value bug is present in L2 v2.3.1 and OFFL v2.4. It has been corrected for RPRO v2.4 and v2.5 and up. The qa-value was corrected in the data going into this study. The recipe to do the correction follows multiple steps with different limits and would be confusing to fully explain here. Therefore we decided to refer to the ATBD where an explanation of the correction is included.

Line 102: What is meant by alternative processors? Please also provide references for that.

An alternative method to estimate uncertainties in the NO2 SCDs is by statistical analysis of the distribution of SCDs corrected for viewing geometry over regions with very little variability in stratospheric NO2. This has been reported in Boersma et al. (2004); Zara et al. (2018), and most recently, for TROPOMI, in van Geffen et al. (2020). We will update the text accordingly.

Line 105: You wrote start fields from TROPOMI? Which data exactly? TM5 has not been introduced yet.

We will introduce TM5 before this statement. We clarify 'starting fields' to 'observed slant columns'.

Line 107: Is Dirksen et al. 2011 a correct reference here? It is a study on OMI stratospheric NO2. It doesn't have any TROPOMI and model differences, which are the topic of this sentence. Is it applied in the TROPOMI or the OMI L2 algorithm?

The Dirksen et al. (2011) reference describes how the data assimilation system to estimate stratospheric NO2 based on NO2 SCDs and TM5 functions. It has been implemented in the same way for TROPOMI as it was for OMI. The O-A and O-F statistics obtained for TROPOMI are consistent with those obtained for OMI as reported in Dirksen et al. (2011). We will include a reference to the TROPOMI L2 ATBD, where it is discussed that the same value was implemented, and which also shows a figure of the O-F statistics in TROPOMI.

Line 109: I can't follow the logic here. You say that you apply a more detailed latitude- and time- dependent L2 uncertainty as derived by Rijsdijk et al. (2024). What is done in Rijsdijk, what is the connection to the sentence before and after?

We extend this to: "*Here, we apply a more detailed latitude- and time-dependent L2 stratospheric uncertainty as derived by Rijsdijk et al. (2025), where the O-F is analysed over latitudinal and day-of-year bands resulting in a look-up-table.*"

Line 113: All these studies are for OMI or GOME-2; are the results applicable to TROPOMI?

Yes. The methods are identical and the TROPOMI instrument is very similar to OMI, so the results are expected to be very similar. We have added this to the text.

Line 133: I think it would be helpful if you are more precise here: "temporal averaged estimates of the column values". Do you mean "of the spatially averaged column values x_o,t", which are, when temporally averaged, named x overline?

We agree this could be clarified. We will change it to: "provide temporal averaged estimates (x overline) of the spatially averaged column values (x0,t)".

Line 135: of the retrieved column (x_i) and its uncertainty (sigma_i)

We agree and will make this change

Line 136: sigma_o,t was not yet introduced, you only mentioned sigma_m,s, and sigma_r,s. I think it would be helpful if you introduce sigm_o,t together in line 131.

We agree and will have this clarified.

Figure 1: Shouldn't it be x_o,t instead of x_o in step1? In step "0" you have x_i, sigma_i, and w_i, but in step 1 and 2 you only have x_o and x_overline, I think it might be helpful to add the uncertainties here as well.

We agree and have made this change.

Line 148: How were the grid resolutions of 0.2°, 0.5° and 1° selected?

The motivation for selecting these resolutions is that many applications such as trend analysis and model evaluations proceed at these resolutions. We also made available L3 data at 2x2.5, the resolution of the global GEOS-Chem model.

Equation 2, Line 153: At this point, I was wondering if sigma_m and sigma_m,s, which you introduced in line 131 together with sigma_r,s are the same. In the following it is more clear that they are not, but it is hard to follow.

We have tried to clarify this by specifying in line 163: "*Each of these ($\sigma_{N_s}$ , $\sigma_{N_s}^{strat}$ , and $\sigma_{M^{tr}}$) are represented by equation 4 [...]*", instead of waiting until after the equation.

Line 180: Are you sure this is the correct reference here, it is not containing TM5 or TROPOMI?

We agree that a better reference here would be Williams et al. (2017). We will change this reference.

Line 186: I think it is important to mention here already that the a priori is in general also a large contribution to the uncertainty but is shown to become irrelevant when the averaging kernel is used (as you mention in line 259) but is still discussed later.

We agree and have made this change.

Line 199: What is the meaning of a correlation length?

The correlation length describes how quickly the correlation decreases the further away two observations are from each other. This can be seen as the typical length over which the correlation reduces by a factor 1/e.

Line 226: How is g determined?

The use of the variable letter 'g' was a mistake on our side. It should have been 'f', the representativeness, which was defined in the previous section.

Line 226: How do you define superobservations?

Superobservations are used interchangeably as 'gridcells in the spatial average'. We will introduce this term on line 224.

Line 233: Is it correct that T is the total number of valid superobservations? I thought that L3 column is averaged over all valid observations, with giving a higher weight to the superobservations.

In the temporal averaging step here, we average over all valid spatial-means (from function 1). 'Superobservation' is a synonym for 'grid cell in the spatial average'. We will make this more clear when the term superobservation is introduced to avoid misunderstandings. A higher weight is then given to superobservations/spatial-means that have a lower representativeness uncertainty.

Line 241/244: You mention the spatial representativeness uncertainty twice but not the a priori uncertainty.

We will change this to also mention the a priori uncertainty

Line 299: Only to avoid misunderstanding, you say "carried out for the NH summer period 1 June -31 August" but the correlation coefficient is determined globally, is this correct? Have you added the information NH to clarify that it is summer on the NH but winter on the SH? Do you expect differences, should you differentiate between NH summer/winter and SH summer/winter?

Yes, 'NH' was added to clarify that it is summer on the NH but winter on the SH. We find a slightly different correlation factor for summer and winter, but decide to go with the highest value to be more conservative in our uncertainty estimate.

Line 321: This is not really clear to me. To correct for what, the temporal representativeness?

The second term in equation 13 is a finite population correction to the normal method to calculate a standard error. Using this finite population correction, the standard error will drop to zero when we have observations available for every single day of the month. We will clarify this in the text by using the term finite population correction.

Line 360: You write "this method" but I think the reference is not clear. Do you mean the sampling test/Wald-Wolfowitz test?

With this method we mean the systematic sampling test. We will clarify this in the text.

Line 384: Probably a combination of longer lifetimes and higher emissions due to heating. Add a reference.

We will add the mention of higher emissions and lifetimes in winter, with a reference to Shah et al. (2020).

Line 389/390: Do you have an idea why the difference in the average L2 relative uncertainty is so large between Amsterdam (52%) and Beijing (28%)?

This is mostly because the actual NO2 columns are higher in Beijing than Amsterdam. As some of the uncertainties are a set value, in general the relative uncertainty is higher for areas with lower tropospheric NO2 columns.

Line 429: I think this reference doesn't contain the 10 mentioned sites, but if it's for SAOZ in general, then mention it directly after SAOZ.

We have moved the reference to directly after SAOZ.

Line 436/Fig 8: You have written an excellent agreement. Do you have an idea why there is perfect agreement from July to October but actually quite some deviations always in spring?

The apparent discrepancy in spring is part of a smoothly varying seasonal bias. The origin is unconfirmed, but our suspicion is that it is related to the use of fixed cross sections in the SAOZ measurements, i.e., these don't take the variation in effective stratospheric temperature into account.

Figure 9: You mention an explanation for the deviation of the Paris site. Do you have an idea for the large deviation of the Dome Concorde site in Antarctica? Is it also visible in the L2 Verhoelst et al. study?

The comparisons at Dome C correspond to difficult validation conditions as the needed solar angles for the SAOZ measurements are only reached during a small part of the year and the scan sequence is often incomplete. Larger deviations are therefore not unexpected.

Figure 10/Line 450: The comparison for Xianghe shows a good agreement in summer but is low biased in the much more polluted winter months. The polluted winter months are the months with higher spatial and temporal variability, which are smeared out in the L3 data. I think this could be discussed in the text.

We mention this effect later, but agree that it is better to discuss it here already.

Figure 9/10/13: Please add data period (2018-2021?).

We will do this in the caption of figure 9. We believe with the caption of the following figures stating 'similar to figure 9….' it is clear enough that it is in the same time period.

Line 493: "clear-sky or low-cloud conditions", I think low-cloud conditions is misleading, please be more precise.

We have changed this to 'low-cloud-fraction conditions'. It refers to L3 grid cells which are partly overcast. Spatial mean is only calculated when at least 30% of the grid cell is covered with valid observations (e.g. clear-sky).

Line 532/Figure A1: Do you have an idea why the difference between the ascending and descending part of the orbits is much less obvious in the Antarctic region?

This has to do with the geometry of the TROPOMI orbit. The offset in the descending part of the orbit is caused by the descending node observations not being included in the data assimilation used to estimate the stratospheric column. In the northern hemisphere, this has a larger effect on pixels west of the equatorial crossing of the orbit. This is because here the latest valid ascending orbit observations included in the data assimilation are made multiple hours ago. In contrast, the portion of the descending node to the east of the equatorial crossing has had recent data-assimilation from a previous orbit (as the orbits move westward). In the geometry of the TROPOMI orbit, the largest portion of the descending mode in the Northern Hemisphere is in this more sensitive part (Figure 1), whereas in the Southern Hemisphere, the largest portion of the descending mode is in the eastern part.

[Figure]

**Figure 1.** Geolocation flags of a single orbit on June 21st 2019.

There are a few other factors compounding this difference. These include higher solar zenith angles in the already sensitive part to the west in the Northern Hemisphere, and a diminishing effect on the uncertainty due to the high surface albedo of the Antarctic Ice Sheet.

**Technical corrections:**

These technical corrections will be made. Thank you for pointing them out.

Line 4: Remove brackets around NO2

Line 6: Introduce Level 2 (L2) like you have done in line 3 for L3 and use L3 and L2 in lines 13 and 14.

Line 14: Replace separate with individual.

Line 14: GCOS is not a commonly known abbreviation, please introduce it.

Line 15: Replace (sub-)columns, I think it is not clear what is meant by that. "Validation of the tropospheric, stratospheric, and total columns"

Line 29: tropospheric NO2 columns instead of tropospheric columns NO2

Line 36: Change measurement techniques to measurements.

Line 38: Change "make them fit for purpose for climate monitoring"
*This correction is unclear to us. What should it be changed into and why is it incorrect?*

Line 42: This is a long sentence, I would suggest splitting it into two sentences: ...by the scientific community. However, L3 data are relevant for model evaluation...

Line 59: A long sentence, you could split it after the reference to Labzovskii.

Line 65: into instead of in to

Line 67: "of with" delete of

Line 68: You mention ESA CCI+ here for the first time; it is not clear what it is. Remove it, or maybe even better, introduce it in your introduction.

Line 80: Change to "are the L2 TROPOMI NO2 tropospheric vertical columns on an orbital basis"

Line 82: Remove brackets to have it like this OMI QA4ECV v1.1 product (Boersma et al., 2018)

Line 82: Add TROPOMI and RPRO or PAL information in front of the v2.3.

Line 83: Split into two sentences: ...(Boersma et al., 2018). These OMI and TROPOMI data products...

Line 84: Add information about which period of data is used. ...which allows for better merging of the datasets, and allows using data from year x to year x.

Line 93: First time use of slant column density, introduce abbreviation SCD or $N_s$. Remove the introduction of SCD in line 100 and use the abbreviation in the following, e.g., line 103.

Line 113: You have used OMI abbreviation already before.

Line 113: Please introduce QA4ECV.

Line 120, 123: Please be precise: stratospheric SCD instead of stratospheric column. Tropospheric vertical column instead of tropospheric column. Abbreviation Nvtrop was not introduced yet.

Line 132: sigma_rs comma is missing between r and s

Line 148: Please correct "we provide spatial the here created dataset at resolution..."

Line 181: I would suggest to mention the used grid sizes earlier: This is coarser than the spatial average grid sizes of 0.2 to 1.0° considered here. Therefore, it is assumed

that the error in the stratospheric column is fully correlated in space with the L3 grid resolution.

Line 218: Change to "the L3 grid resolution"

Figure 2: "Larger black points" instead of "Larger points"

Line 269/270: an/the albedo

Line 299: Add: for the NH winter period 1 January to 31 March

Line 399/Figure 5/Figure 6: Add that it is the vertical column.

Line420: remote sensing measurements

Line 425: L3 qa_value

Line 432: Replace S5P with TROPOMI, you have always used TROPOMI.

Line 433: Delete operator.

Line 443: Various instruments/operators instead of sources?

Line 449: The location of the given reference doesn't make sense here.

Line 451: Please check this sentence for typos.

Figure 12: Please add the L2 version, similar as in Figure 10.

Line 477: I think it is -39% instead of -35%

Line 480: from nearly 20%